# INHERITUNE: TRAINING SMALLER YET MORE ATTENTIVE LANGUAGE MODELS

## ABSTRACT

Large Language Models (LLMs) have achieved remarkable performance across various natural language processing tasks, primarily due to the transformer architecture and its self-attention mechanism. However, we observe that in standard decoder-style LLMs attention matrices degenerate to single-column for deeper layers. Layers in this state unable to learn anything meaningful and mostly redundant; we refer to these as lazy layers. The goal of this paper is to train smaller models by eliminating this structural inefficiency without compromising performance.

Motivated by this observation, we propose **Inheritune**, a simple yet effective training recipe for developing smaller, high-performing language models. Smaller models trained with Inheritune inherits early transformer layers from a larger pretrained model, then retrains and progressively expands the smaller model until it matches or exceeds the performance of the larger model. We demonstrate that Inheritune enables the training of various sizes of GPT-2 models on datasets like OpenWebText-9B and FineWeb_Edu. Models trained with Inheritune, despite having significantly fewer layers, match or even surpass the performance of their larger counterparts. For instance, our 16-layer GPT-2 medium variant achieves comparable performance to the standard 24-layer GPT-2 medium model.

## 1 INTRODUCTION

Large Language Models (LLMs) are built with decoder-style transformer blocks (Vaswani et al., 2017). These models are typically designed to be large, with a significant portion of their parameters dedicated to their depth, with multiple transformer blocks stacked with eachother building model capacity. Each block or layer in the stack refines the representations learned by the previous blocks, allowing the model to develop a nuanced understanding of the input data. As these models scale in depth and size, their performance tends to improve Kaplan et al. (2020); Hoffmann et al. (2022), benefiting from increased model capacity.

The causal self-attention (hereafter referred to as attention) mechanism is arguably the most crucial component of a transformer block. It allows models to combine tokens as a weighted linear sum of their attention scores, effectively capturing long-range dependencies and contextual relationships within text data. However, as models grow in depth, they often encounter a phenomenon known as attention degeneration caused by collapse in the attention rank ((Noci et al., 2022; Dong et al., 2021; He et al., 2023)). Notably, this phenomenon has not been studied in the context of standard LLMs. A formal discussion on attention degeneration is provided in Section 2.

In this paper, we empirically analyze 24-layer GPT-2 medium and 36-layer GPT-2 large models (decoder-style LLMs) Radford et al. (2019) for attention degeneration and observe that many deeper layers in both models exhibit rank-1 attention matrices. Further investigation reveals that most of these rank-1 matrices are also single-column, i.e. their mass is concentrated to a single column. Our attention matrix analysis is presented in Figure 1. We term these deeper layers, where all attention matrices of a given layer are degenerated, as *lazy layers*.

Motivated by this new finding we aim to develop performant small base language models (LMs) utilizing weights from in-efficient larger base LMs. A base LM is a decoder-style model trained solely for next-token prediction without additional enhancements like instruction tuning or reinforcement learning with human feedback (RLHF). Our proposal is straightforward, we start by initializing our

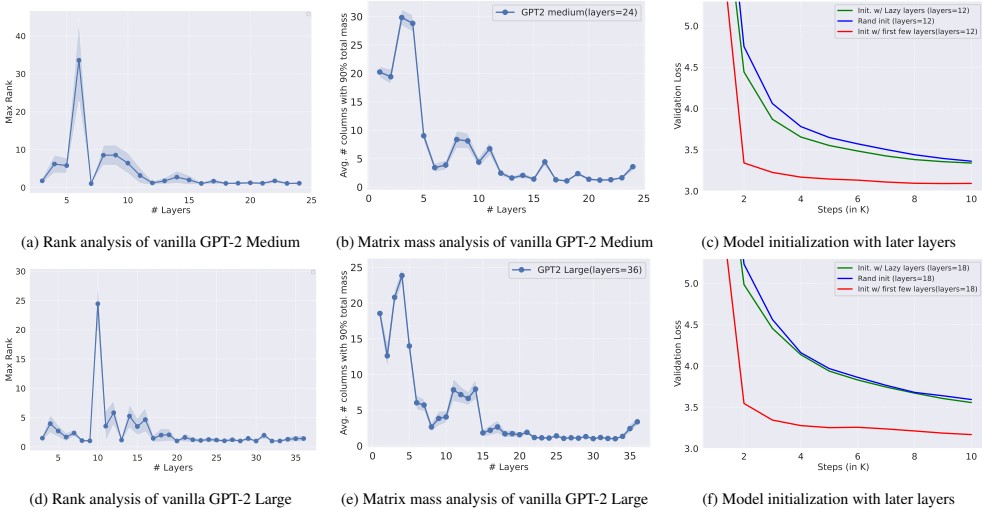

(a) Rank analysis of vanilla GPT-2 Medium  (b) Matrix mass analysis of vanilla GPT-2 Medium  (c) Model initialization with later layers

(d) Rank analysis of vanilla GPT-2 Large  (e) Matrix mass analysis of vanilla GPT-2 Large  (f) Model initialization with later layers

Figure 1: **Attention matrices of many deeper layers often degenerates to single-column matrices in regular decoder style LLMs, layers with fully degenerated attention fails to learn meaningful representations.** We computed a single attention matrix with 100 tokens from the OpenWebText validation set with 4M tokens. Next we performed 100 runs and plotted the mean and std of the max rank and mass as a function of layers for our rank and mass analysis respectively. Figure (a and d): An analysis of a 24-layer GPT-2 medium and a 36-layer GPT-2 large shows the max rank of the attention matrices across all layers. Figure (b and e): A closer look at the the same GPT-2 models also reveals that the dominant mass proportion of several attention matrices is concentrated in a single-column particularly in deeper layers. Figure (c and f): When initializing 12-layer and 18-layer variants[1] of the vanilla GPT-2 medium and GPT-2 large models with deeper layers (*Lazy layers*) exhibiting degenerated attention–their performance is comparable to models with random initialization. However, initializing models with early layers leads to significantly better generalization and convergence.

smaller LM (target) using the first few blocks from a large pre-trained LM (reference). We then train the target model for a specified number of steps. After this initial training, we incrementally grow the target model by adding more blocks, continuing the training process until it matches or surpasses the pre-train validation loss (also val loss) of the reference model. During the growth phase, the newly added blocks can be initialized with *lazy layers* of the reference LM. We refer to this simple yet effective training approach as Inheritune.

In summary, our key contributions are as follows:

1. **Analysis of Attention Degeneration Leading to Lazy Layers.** We empirically investigate attention degeneration in standard LLM settings. Our analysis shows that rank-collapsed attention matrices often exhibit single-column structures, revealing a significant structural inefficiency in the attention mechanism of standard LLMs in deeper layers (see Figure 1 and Figure 2).

2. **Introduction of Inheritune.** We propose Inheritune as an approach to effectively train high-performing, smaller models. This method involves inheriting a few early blocks from a larger pre-trained model and progressively growing and training the smaller model. The initialization is entirely zero-shot. We validate the effectiveness of Inheritune through comprehensive experiments using GPT-2 xlarge (1.5B), GPT-2 large (770M), and GPT-2 medium (355M) models, trained on the OpenWebText dataset with 9B tokens (with data repetition) and the FineWeb_edu dataset with 100B tokens (without data repetition).

3. **Evaluation Against Multiple Baselines.** Models derived using Inheritune consistently outperform various baselines, including much larger models trained from scratch (refer Table 1), model initialization and efficient training baselines (refer Table 2), and models

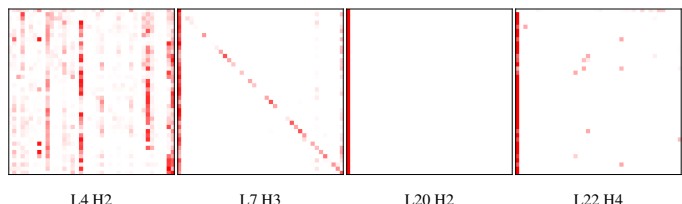

L4 H2          L7 H3          L20 H2          L22 H4

24 layer GPT2 medium model trained from scratch.

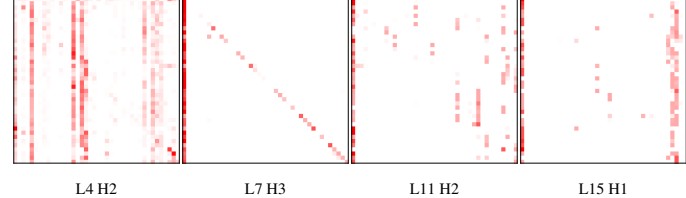

L4 H2          L7 H3          L11 H2          L15 H1

Our 16-layer GPT2 medium model trained using Inheritune.

Figure 2: Inheritune **preserves effective attention patterns in smaller models.** Comparison of attention patterns across layers (L) and heads (H) in two GPT2-medium models: (top) 24-layer model trained from scratch, (bottom) 16-layer model trained with Inheritune. Attention maps are averaged over three randomly selected string, with 40 tokens each from the validation. Darker colors indicate higher attention scores. Inheritune maintains focused attention even in deeper layers, contrasting with the uniform patterns in the standard model's later layers.

trained using two knowledge distillation techniques (refer Figure 3). In settings where training tokens are not repeated, we observe similar trends (refer Figure 4).

## 2 ATTENTION DEGENERATION IN STANDARD DECODER-STYLE LLMS

**Preliminaries:** A vanilla transformer-based model consists of $L$ transformer blocks (layers). The model operates on an input sequence $X \in \mathbb{R}^{T \times d}$, where $T$ denotes the sequence length (number of tokens), and $d$ represents the embedding dimension or model hidden size. The output of each layer $l$ is denoted as $X^{(l)} \in \mathbb{R}^{T \times d}$.

Each transformer block primarily consists of two sub-layers: a self-attention block and a position-wise feed-forward network (FFN). The self-attention mechanism enables the model to weight the relevance of different tokens in the sequence relative to each other. Specifically, for a single attention head, the attention computation is defined as $\text{Attention}(Q, K, V) = \underbrace{\text{softmax}\left(\dfrac{QK^{\top}}{\sqrt{d_k}}\right)}_{\textbf{Attention matrix: } A(X)} V$

where the queries $Q = XW_Q$, keys $K = XW_K$, and values $V = XW_V$ are linear transformations of the input $X$. Here, $W_Q, W_K \in \mathbb{R}^{d \times d_k}$ and $W_V \in \mathbb{R}^{d \times d_v}$ are the weight matrices for the queries, keys, and values, respectively. Typically, $d_k = d_v = \frac{d}{h}$, where $h$ is the number of attention heads. In this single-head scenario, we set $d_k = d_v = d$.

The **attention matrix** $A(X) \in \mathbb{R}^{T \times T}$ captures the pairwise attention scores between all token positions in the sequence. The softmax is applied row-wise. The attention matrix $A(X)$ is then used to compute a weighted sum of the value vectors.

Previous research by Dong et al. (2021) and He et al. (2023) has shown that in self-attention networks (SANs) without residual connections and feed-forward networks (FFNs), the rank of an attention matrix converges to rank-1 doubly exponentially with respect to the depth of the model. This phenomenon, known as rank collapse of attention matrices, results in a loss of expressive power as the attention mechanism attends to all tokens uniformly. Noci et al. (2022) showed that even with

residual connections (without layernorm) attention matrices can still loose rank in deeper layers if the residual connections are not scaled by $1/\sqrt{L}$. Interestingly they also linked the rank collapse to vanishing gradients of the keys and queries in deeper layers which affects the overall trainability of the transformer based models. However, these findings do not directly apply to the standard LLMs, as transformer blocks in these models include residual connections, layernorms and FFNs, which are expected to mitigate both rank collapse and the vanishing gradient problem.

**Approximate Rank Computation of Attention Matrices**    In this paper, we deeply analyzed the structure of attention matrices to diagnose the presence of rank collapse or similar phenomena in standard transformer-based LLMs using GPT-2 models. For our first analysis, we compute the approximate rank (referred to as rank hereafter) of $A(X)$ for all attention heads within each layer. Formally, we began by computing the Singular Value Decomposition (SVD) of $A(X) = U\Sigma V^\top$, where the diagonal entries $\{\sigma_i\}_{i=1}^T$ of $\Sigma$ represent the singular values, quantifying the variance captured by each corresponding singular vector of $A(X)$. To determine the minimal number of singular values required to capture 90% of the total variance, we solved: $k^* = \min\left\{k \in \{1, 2, \ldots, T\} \mid \frac{\sum_{i=1}^k \sigma_i^2}{\sum_{j=1}^T \sigma_j^2} \geq 0.90\right\}$. Here, $k^*$ is the approximate rank of $A(X)$ computed using the explained variance method.

**Dominant Single-Column Structure in Attention Matrices**    We further investigated the dominant structure of these rank-1 attention matrices and observed that, on an average, many of these matrices have their mass concentrated in a single column. This intrinsic structure can be viewed as a special case of rank-1 attention matrices. To quantify this, we computed the proportion of the matrix mass contributed by each column $j$ of $A(X)$ by computing $\frac{\|A_{\cdot,j}\|_2^2}{\|A(X)\|_F^2}$, where $A_{\cdot,j}$ denotes the $j$-th column of $A(X)$, $\|A_{\cdot,j}\|_2$ is the $\ell_2$-norm of that column, and $\|A(X)\|_F$ is the Frobenius norm of $A(X)$. Next to determine the minimal number of columns required to capture 90% of the total mass we solved; $m^* = \min\left\{m \in \{1, 2, \ldots, T\} \mid \sum_{j=1}^m \frac{\|A_{\cdot,j}\|_2^2}{\|A(X)\|_F^2} \geq 0.90\right\}$. Here, $m^*$ represents the minimal number of columns needed for the cumulative column mass ratios to reach or exceed 90%.

**Degeneration of Attention Matrices in GPT-2 Models**    In Figure 1, we present the layer-wise analysis of rank and mass. For this analysis, we computed $A(X)$ using 100 randomly selected samples from the validation set of OpenWebText valset with 4M tokens, each with a sequence length of $T = 100$ tokens, across all attention heads within each layer. Next we computed the rank with 90% variance threshold and for every layer we chose the maximum rank across all the heads. In Figure 1a and 1d we plotted maximum rank as a function of layers for 100 runs with mean and standard deviation (std); it's quite evident that many deeper layers exhibit all rank-1 attention matrices. A rank-1 $A(X)$ has a $2T - 1$ degrees of freedom i.e. expressive power compared to a full rank $A(X)$ which is $T^2$. We highlight that this rank collapse is happening for both GPT-2 medium and GPT-2 Large models with skip connections and FFNs, extending the findings of Dong et al. (2021) and Noci et al. (2022) to standard LLM architectures. Next based on our mass analysis we demonstrate that most of the rank collapsed attention matrices are also single-column matrices as depicted in Figure 1b and Figure 1e. We follow the protocol of analysis as discussed in the rank analysis. A single-column $A(X)$ has an expressive power of $T$ i.e. $1/T$ times compared to a full rank $A(X)$. This degeneration of attention matrices in deeper layers provides quantitative evidence for the existence of *lazy layers*. Specifically, we observe that some deeper layers exhibit complete degeneration of all attention matrices across all attention heads, indicating reduced performance and less effective token mixing.

**How much transferable knowledge should these *lazy layers* hold compared to their earlier counterparts?**    To answer this question, we initialized a 12-layer GPT-2 medium variant[2] and an 18-layer variant of GPT-2 large using lazy layers extracted from pre-trained 24-layer GPT-2 medium and 36-layer GPT-2 large models. These pre-trained models are trained on the OpenWebText-9B dataset for 100K steps. We then fine-tuned these GPT-2 variants on the same dataset for an additional 10K steps. For comparison, we conducted two baseline experiments where the GPT-2 variants were initialized either with the first few transformer blocks or with random initialization. As shown in

---

[2]A variant shares the same configurations as the parent model but has fewer layers.

Figures 1c and 1f, models initialized with lazy layers demonstrate poor transferability, performing similarly to models with random initialization. This provide additional evidence that lazy layers with fully degenerated attention, fails to learn meaningful representations.

## 2.1 ATTENTION PATTERN VISUALIZATION

To provide further evidence of lazy layers and provide a preview of our solution, we visualized attention patterns across various layers of a vanilla 24-layer GPT-2 medium model. Fig. 2 shows the attention patterns for both a vanilla 24-layer model trained from scratch and a 16-layer model trained using our proposed method, Inheritune. Note just for the sake of better visualization we visualized full attention and not causal attention, in practice GPT-2 models computes causal attention. We computed these attention matrices using randomly selected strings from the val set of OpenWebText and took 40 tokens averaged over 3 runs.

In the 24-layer model trained from scratch (top row of Fig. 2), we observe a clear progression in attention patterns. The early layers (L4 and L7) exhibit structured patterns with a mix of local and global attention Gong et al. (2019); Beltagy et al. (2020); Chen et al. (2021). In contrast, the deeper layers (L20 and L22) display more uniform patterns, indicating a loss of focus. This uniformity is a hallmark of *lazy layers*, where the attention mechanism loses its ability to selectively focus on specific relevant tokens. In contrast, our 16-layer model trained with Inheritune (bottom row) demonstrates more focused and effective attention patterns, even in its later layers (L11 and L15). This striking difference suggests that our method makes model more attentive and addresses attention degeneration, potentially

---

**Algorithm 1** Inheritune: Training Recipe for Small Language Models

**Require:** Reference model $\mathcal{M}_{\text{ref}}$ with $k$ layers, datasets $\mathcal{D}_{\text{train}}$ and $\mathcal{D}_{\text{val}}$, steps T
1: Initialize $\mathcal{M}_{\text{tgt}}$ with first $n = k/2$ layers from $\mathcal{M}_{\text{ref}}$
2: Train $\mathcal{M}_{\text{tgt}}$ on $\mathcal{D}_{\text{train}}$ for T steps
3: **while** $\mathcal{M}_{\text{tgt}}$ performance $< \mathcal{M}_{\text{ref}}$ performance on $\mathcal{D}_{\text{val}}$ **do**
4:     Grow $\mathcal{M}_{\text{tgt}}$ by inheriting additional layers
5:     Train $\mathcal{M}_{\text{tgt}}$ for T steps
6: **end while**
7: **return** Optimized model $\mathcal{M}_{\text{tgt}}$

---

leading to more efficient models in compact size (also refer Figure 9 and Figure 10). We will discuss these results in more detail after introducing our method, but this preview underscores the promise of our approach.

## 3 INHERITUNE: OUR PROPOSED TRAINING RECIPE

This section offers a detailed description of our method, key implementation considerations, and how it addresses the inefficiencies present in current architectures.

Recall we have established the issue of attention degeneration with two motivating examples, highlighting specific inefficiencies in pre-trained LLMs. In this paper, we turn this challenge into an opportunity to create smaller base LMs that are equally performant, achieving similar or lower validation loss compared to their larger, less efficient counterparts, which we refer to as reference models. Our proposed solution builds on two key insights: (1) the early layers of deep LLMs provide effective model initialization, and (2) multiple lazy layers can be collapsed into fewer layers and re-trained to improve the model capacity.

**Setup:** We split the dataset into a training set $\mathcal{D}_{\text{train}}$ and a validation subset $\mathcal{D}_{\text{val}}$. Next, we assume that there exists a pre-trained reference model $\mathcal{M}_{\text{ref}}$, comprising $k$ layers, represented by $W_{\text{ref}} = \{W_0, W_1, \ldots, W_{k-1}\}$ trained with $\mathcal{D}_{\text{train}}$ for T steps. We want to train a smaller model $\mathcal{M}_{\text{tgt}}$ with the same or better validation loss (lower is better) compared to its larger counterpart $\mathcal{M}_{\text{ref}}$.

We now present Inheritune, our proposed training recipe for efficiently developing small base language models (LMs). Inheritune operates on the principle of zero-shot initialization and progressive growth. The Inheritune process consists of three main steps, which we present below and formalize in Algorithm 1:

1. **Inherit:** Initialize $\mathcal{M}_{\text{tgt}}$ with the first $n = k/2$ layers of $\mathcal{M}_{\text{ref}}$, including weights, prediction head, and token embeddings.

2. **Train:** Train $\mathcal{M}_{\text{tgt}}$ for T steps on $\mathcal{D}_{\text{train}}$ and evaluate on $\mathcal{D}_{\text{val}}$.

3. **Grow:** If needed, increase $\mathcal{M}_{\text{tgt}}$'s size and repeat steps 1-2 until desired performance is achieved.

With our method now formally described, we turn to empirical validation. In the following sections, we present comprehensive results demonstrating Inheritune's effectiveness across various scenarios, including different model sizes and data regimes. In addition, we conducted an in-depth ablation study to analyze the impact of initialization on performance, providing insights into the adaptability of our approach.

## 4 EXPERIMENTS

We evaluate Inheritune through a series of comprehensive experiments using GPT-2 xlarge (1.5B), GPT-2 large (770M) and GPT-2 medium (355M) models, Radford et al. (2019) pre-trained on the 9B tokens OpenWebText dataset (Gokaslan & Cohen, 2019). These models are trained with data repetition, meaning data is randomly sampled with replacement during batch creation. This experimental setup is adapted from Liu et al. (2023); Sanyal et al. (2024). For evaluation we compare model(s) trained with Inheritune with baselines from three key settings: a) baseline models trained from scratch with random initialization, b) baseline Models trained using various zero-shot initialization techniques c) baseline models trained with knowledge distillation. Table 10 provides detailed specifications for all models used in our experiments. Finally, we conduct a thorough ablation study of our initialization strategy, focusing on 16-layer GPT-2 medium variant(s).

We provide experimental details our Inheritune training recipe using a GPT-2 large model as an example; similar procedure was applied to train other models. Our methodology for applying Inheritune involves the following steps:

1. **Reference Model:** We train the full 36-layer GPT-2 Large model on $\mathcal{D}_{\text{train}}$ for 100K steps and evaluate its validation loss ( log-perplexity) on $\mathcal{D}_{\text{val}}$. This establishes our benchmark validation loss.

2. **Model initialization** We initialize an 18-layer model ($n = k/2$) using the trained 36-layer model as reference.

3. **Training and Evaluation:** We train the 18-layer model on $\mathcal{D}_{\text{train}}$ for $T$ steps and evaluate its validation loss.

4. **Iterative Refinement:** If the smaller model's performance is inferior, we incrementally increase its size by two layers and repeat steps 2-3 until we achieve parity with the reference model's validation loss.

**Baselines trained from scratch (rand init.) :**   We compare our Inheritune-derived model against much larger GPT-2 reference models trained from scratch for the same number of steps and similar-sized models trained from scratch for both the same and double the number of training steps.

**Baselines trained with various model initialization and efficient training techniques.**   Here we compare our model derived using Inheritune, to similar sized models trained with various zeroshot model initialization and effcient training techniques such as stacking, hybrid stacking, and half-width initialization. We explain these baseline training recipes using GPT-2 large and its variants as an example and apply the same process for other models.

**Stacking** Gong et al. (2019); J. Reddi et al. (2023) is a model initialization and efficient (stage-wise) training recipe. We train a 9-layer GPT-2 large variant from scratch for 100K steps, then expanded the model to 18 layers by copying the weights from layers 0-8 to layers 9-17. Finally we re-trained this new 18-layer GPT-2 large variant, using stacking initialization for an additional 100K steps.

**Hybrid stacking**: Hybrid stacking is stacking but utilizes a large pre-trained reference model for initialization instead of using its own pre-trained weights. We took the weights of layers 0-8 from the reference 36-layer GPT-2 large model and expanded it to a 18-layer model by copying the weights to layers 0-17. We then trained this new 18-layer GPT-2 variant for 100K steps.

| Models | Layers | Initialization | Steps | Pre-train | Downstream (0-shot) | |
|---|---|---|---|---|---|---|
| | | | | Val loss ($\downarrow$) | Wikitext ($\downarrow$) | Lambada |
| | 24 | rand init | 100K | 2.81 | 31.93 | 36.54 |
| | 16 | rand init | 100K | 2.86 | 33.67 | 34.60 |
| | 16 | rand init | 200K | 2.83 | – | – |
| GPT-2 Medium | 12 | Ours | 100K | 2.87 | – | – |
| | 14 | Ours | 100K | 2.84 | – | – |
| Final Model $\longrightarrow$ | 16 | **Ours** | 100K | **2.81** | 32.04 | 35.96 |
| | 36 | rand init | 100K | 2.85 | 34.84 | 34.14 |
| GPT-2 Large | 18 | rand init | 100K | 2.97 | 37.63 | 30.97 |
| | 18 | rand init | 200K | 2.84 | – | – |
| | 18 | **Ours** | 100K | **2.80** | 35.38 | **34.64** |
| | 48 | rand init | 100K | 2.65 | 25.45 | 39.90 |
| GPT-2 xLarge | 24 | rand init | 100K | 2.69 | 28.32 | 38.46 |
| | 24 | rand init | 200K | 2.62 | – | – |
| | 24 | **Ours** | 100K | **2.64** | 25.52 | **43.30** |

Table 1: **Inheritune achieves superior performance with reduced model size.** Comparison of Inheritune-trained models (24-layer GPT-2 xLarge, 18-layer GPT-2 Large, 16-layer GPT-2 Medium) against full-sized counterparts and extended training baselines. Metrics include pre-training validation loss ($\downarrow$), zero-shot Wikitext ($\downarrow$) and Lambada performance. Note: GPT-2 Large and xLarge took one round of training; GPT-2 Medium took three rounds.

| Models | Layers | Recipe | Steps | Pre-train Val loss ($\downarrow$) |
|---|---|---|---|---|
| | 24 | half-width | 100K | 3.04 |
| | 16 | stacking | 100K | 2.84 |
| GPT-2 Medium | 16 | hybrid-stacking | 100K | 2.83 |
| | 16 | **Ours** | 100K | **2.81** |
| | 36 | half-width | 100K | 3.06 |
| | 18 | stacking | 100K | 2.87 |
| GPT-2 Large | 18 | hybrid-stacking | 100K | 2.89 |
| | 18 | **Ours** | 100K | **2.80** |
| | 48 | half-width | 100K | 2.77 |
| | 24 | stacking | 100K | 2.65 |
| GPT-2 xLarge | 24 | hybrid-stacking | 100K | 2.64 |
| | 24 | **Ours** | 100K | **2.64** |

Table 2: Inheritune **outperforms baseline zero-shot initialization and efficient training techniques.** Comparison of pre-training validation loss for GPT-2 xLarge, GPT-2 Large and GPT-2 Medium variants. Inheritune-derived models consistently achieve lower loss compared to models initialized with stacking, hybrid stacking, and half-width techniques.

**Half width**: We initialized the baseline GPT-2 large variant across the width dimension and preserved the entire depth. We copied the weights of the first half the attention heads (0-9) and MLPs of the GPT-2 large reference model into baseline GPT-2 variant with half the width but all layers.

**Baselines trained with Knowledge Distillation** As a baseline, we first apply logit-based knowledge distillation Hinton et al. (2015) to train a 16-layer GPT-2 medium variant (student) initialized randomly. For the second baseline, we use a DistillBERT-style approach Sanh et al. (2019), where the student model 0-11 layers are initialized with every alternate block of its teacher, and the remaining 4 blocks are initialized using layers 18, 19, 20, and 21 of the teacher. Both baselines are trained for 14K steps, using a vanilla 24-layer GPT-2 medium model as the teacher (our reference model).

## 4.1 RESULTS AND DISCUSSIONS

**Models trained with Inheritune outperforms much larger models trained from scratch.** We present our main results in Table 1. Our 24-layer, 18-layer, and 16-layer variants derived using Inheritune from the vanilla 48-layer GPT-2 xlarge, 36-layer GPT-2 large, and 24-layer GPT-2 medium achieve comparable or lower validation losses than their full-sized counterparts when trained for the same number of steps (100K steps). Our GPT-2 xlarge and GPT-2 large variants undergo one round of Inheritune training, while for GPT-2 medium, we perform three rounds of training with 12, 14, and 16 layers. We also evaluate all the models on two next-word prediction downstream tasks in a zero-shot setting using the Wikitext Merity et al. (2016) and Lambada Paperno et al. (2016) datasets. The downstream performance of our GPT-2 models derived using the Inheritune recipe matches their much larger reference models. From the convergence perspective, some prior works have made connections between over-parameterization and faster convergence Bengio et al. (2005); Vaswani et al. (2018). In supplementary Figure 5, we show that the small LMs derived with Inheritune, although smaller compared to their reference models, still converge as fast as their large-size reference models.

**Models trained with Inheritune outperform same-sized models trained from scratch.** Table 1 demonstrates that GPT-2 variants trained with Inheritune outperform their same-sized counterparts trained from scratch, both when trained for the same number of steps and even when trained for double the steps (200K). This result underscores the efficiency of our approach. The only exception is the 24-layer GPT-2 xlarge variant, which surpasses both our model and the full-sized model when trained for 200K steps.

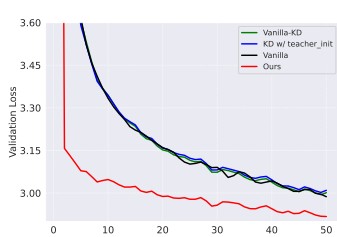

**Models trained with Inheritune outperform all zero-shot model initialization baselines.** In Table 2, we compare GPT-2 xlarge, GPT-2 large, and GPT-2 medium variants trained with Inheritune against same-sized variants trained with stacking, hybrid, and half-width initialization baselines. The half-width baseline performs poorly, revealing the limitations of naive width reduction. While stacking and hybrid stacking demonstrate reasonable performance, they still fall short compared to Inheritune. Across all cases, Inheritune consistently outperforms these baselines, highlighting its effectiveness as an initialization strategy. For a detailed view of the training dynamics across all methods, refer to the training curves in supplementary Figure 6.

Figure 3: **A 16-Layer GPT-2 medium variant derived using Inheritune converges faster and generalizes better than a same-sized model trained with Logit-based distillation baselines.** We conducted vanilla KD Hinton et al. (2015) and DistillBERT-style KD Sanh et al. (2019) with teacher initialization, using a 24-layer GPT-2 medium as the teacher for both KD baselines.

**Distillation vs Inheritune.** In Figure 3, we compare 16-layer GPT-2 medium variants derived using vanilla knowledge distillation Hinton et al. (2015) and DistillBERT-style distillation Sanh et al. (2019), which leverages teacher layers for model initialization, vanilla training from scratch and method. Our model demonstrates faster convergence and significantly better final generalization after 50K steps. Additional distillation experiments can be found in the supplementary materials.

## 4.2 ABLATIONS

We conducted extensive experiments to better understand which sub-module initializations within a transformer block lead to improved generalization (in terms of validation loss) and faster convergence. For these ablations, we fixed the model to a 16-layer GPT-2 medium variant and explored three different sub-module initializations using weights from a 24-layer GPT-2 medium reference model. We initialize the transformer blocks with 1) attention ((key, query, value, and projection) and the layernorm[3] weights (attn w/ layernorm), 2) attention and mlp weights without the layer-norm (attn+mlp w/o layernorm), and 3) mlp weights with the layer norm (mlp w/ layernorm). We emphasize that Inheritune performs initialization by inheriting attention and mlp weights with the layer norm (attn+mlp w/ layernorm).

---

[3]In GPT-2 models layernorm blocks are parameterized.

| Layers | Initialization | Steps | Val loss (↓) |
|--------|----------------|-------|--------------|
| 16 | attn (w/ layernorm) | 100K | 2.84 |
| 16 | mlp (w/ layernorm) | 100K | 2.85 |
| 16 | attn+mlp (w/o layernorm) | 100K | **2.80** |
| 16 | **Ours** (attn+mlp w/ layernorm) | 100K | **2.81** |

Table 3: **Impact of initializing various sub-modules within a transformer block.** We compare validation loss of a 16-layer GPT-2 medium variant when different sets of sub-modules are initialized with weights from the first 16 layers of a 24-layer GPT-2 medium reference model. All models are trained on the OpenWebText-9B dataset. Key findings: (1) Inheritune initialization and attention + MLP initialization result in similar performance improvements; (2) layernorm initialization shows minimal impact. A detailed training curve is shown in Figure 7.

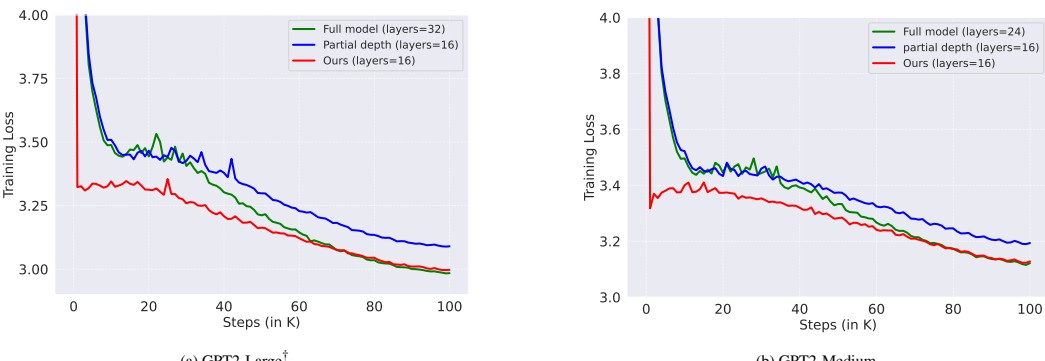

(a) GPT2-Large[†]                                  (b) GPT2-Medium

Figure 4: **Models derived using** Inheritune **without data repetition converges faster and matches the final validation loss of the full-sized model despite using lesser layers.** Additionally, the model trained using Inheritune demonstrates data efficiency, achieving a lower validation loss in fewer steps compared to its full-sized and half-sized counterparts until 80% of the training.

As shown in Table 3, models trained with attention and mlp weights demonstrated the best performance, regardless of the layer norm initialization. A detailed validation loss vs training steps plot is presented in supplementary Figure 7. We conclude that initializing both attention and MLP weights provides a clear advantage. Surprisingly, we also observed that initializing either the attention or mlp weights resulted in similar improvements in both convergence speed and final validation loss.

## 5 TRAINING WITHOUT DATA REPETITION

**Are the gains we observe due Inheritune recipe is merely a consequence of over-fitting due to data repetition?** To investigate this, we conducted additional training experiments without data repetition, following standard LLM pre-training practices as discussed in Touvron et al. (2023a); Biderman et al. (2023). Moreover, we utilized a high-quality pre-training dataset, **Fineweb_edu** Penedo et al. (2024), which contains 100B tokens and has been deduplicated and filtered to ensure high data quality.

We trained a 32-layer GPT-2 large[†] (668M) and a 24-layer GPT-2 medium (355M) reference model from scratch. Next, we trained two 16-layer variants: one derived from GPT-2 large[†] and the other from GPT-2 medium, using their respective reference models following Algorithm 1. Finally, we trained baseline 16-layer variants from scratch for comparison. All these models are trained for 100K steps. The model configurations and training hyper-parameters can be found in supplementary material.

As shown in Figure 4, our GPT-2 variants trained using Inheritune consistently perform on par with their full-sized counterparts and outperform their same-sized counterparts in terms of training loss. In LLM pre-training literature where data is not repeated, training loss has been shown to be a reliable metric Touvron et al. (2023a;b). Additionally, we conducted zero-shot downstream evaluations using

lm-evaluation-harness Gao et al. (2024) on a variety of tasks, including ARC-easy (ARCE; Clark et al. (2018)), LAMBADA Paperno et al. (2016), SciQ Welbl et al. (2017), Hellaswag Zellers et al. (2019), and PIQA Bisk et al. (2020). As shown in Table 4 on an average models demonstrate superior performance than baseline models trained from scratch.

| Models | Recipe | Layers | ARCE (acc) | PIQA (acc) | SciQ (acc) | Hellaswag (acc norm) | Lambada (acc) | **Average** |
|---|---|---|---|---|---|---|---|---|
| | rand init | 24 | 51.05 | 61.81 | 74.8 | 30.79 | 20.28 | 47.74 |
| GPT-2 Medium | rand init | 16 | 49.92 | 61.92 | 73.3 | 29.56 | 19.54 | 46.84 |
| | **Ours** | 16 | 51.26 | 61.81 | 73.8 | 30.55 | 23 | **48.08** |
| | rand init | 32 | 52.48 | 64.58 | 75.3 | 32.65 | 22.2 | 49.44 |
| GPT-2 Large[†] | rand init | 16 | 50.34 | 63.11 | 75 | 30.86 | 21.56 | 48.17 |
| | **Ours** | 16 | 52.9 | 63.55 | 76.1 | 32.14 | 24.06 | **49.75** |

Table 4: **Models trained with Inheritune outperforms both their larger and same-size counterparts trained from scratch on average zero-shot downstream performance.** For evaluation we choose accuracy (acc) and normalized accuracy (acc norm) metrics following Open LLM leaderboard Beeching et al. (2023). All the models are trained with FineWeb_edu.

## 6 RELATED WORKS

**Attention degeneration** has been studied in the past through the lens of attention rank collapse Dong et al. (2021) leading to representation collapse, and attention entropy collapse Zhai et al. (2023) leading training instability. This also has been studied is a theoretical setup for transformer models by Noci et al. (2022); Barbero et al. (2024). Recently He et al. (2023) address rank collapse in self-attention networks (SANs) without residual connections or layer norms, using two model initialization techniques that enable faithful signal propagation—i.e., $\Sigma_L$ of $A(X^L)$ does not collapse in deeper layers. However, this approach significantly slows down training. Noci et al. (2022) proposes scaling residual connections by $1/\sqrt{L}$, while Barbero et al. (2024) suggest that adding additional tokens to already long sequences of repeated tokens can help mitigate degeneration. In contrast to prior works, we address attention degeneration by developing smaller models that eliminate structural inefficiencies and training these models to match the performance of their larger, inefficient counterparts.

**LLM training recipes and model initialization.** The stacking method Gong et al. (2019); J. Reddi et al. (2023) employs a stage-wise training strategy that uses weights from initial layers to initialize later layers has been shown to be effective for LLM training both empirically Gong et al. (2019); J. Reddi et al. (2023); Du et al. (2024) and theoretically Agarwal et al. (2024). Knowledge distillation Hinton et al. (2015) has been very successful in training small LMs in some cases Turc et al. (2020); Sanh et al. (2019) the smaller student model is also initialized with teacher layers-though this is often done without clear explanation or intuition. Recent works in model initialization, such as Trockman & Kolter (2023), have studied synthetic attention patterns for initialization, primarily in vision settings. However, such methods have limited success in language models. Xu et al. (2024) use weight initialization for faster fine-tuning of vision models. In contrast, our proposed recipe focuses on creating smaller model by eliminating specific structural inefficiency in *lazy layers*. This distinction sets our work apart in terms of both objective and methodology.

## 7 CONCLUSION

In this paper, we identified a structural flaw in the attention mechanism of deep decoder-style LLMs, where many deeper layers tend to lose rank and converge into single-column matrices. To address this, we propose Inheritune, to train smaller models that inherits early blocks from a larger model and expands the architecture gradually, matching the performance of the reference model. We validated Inheritune on GPT-2 models of varying sizes, achieving efficient smaller models without performance loss on the OpenWebText-9B and FineWeb_edu datasets.

## 8 REPRODUCIBILITY STATEMENT

To promote reproducibility within the research community, we have provided our complete codebase in a compressed ZIP format. Additionally, we offer a detailed description of all hyperparameters used in our experiments. These resources are intended to enable other researchers to accurately replicate our study and verify our results.

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

# Supplementary Materials

CONTENTS

## A    SUPPLEMENTARY EXPERIMENTS

We provide additional training plots for our main results discussed in Section 4.1 as shown in Figure 5 and Figure 6. In Figure 5 (also refer Table 1 we compare our GPT-2 variants with baseline models trained form scratch. In Figure 6 (also refer Table 2) we compare our GPT-2 variants with baseline models trained using baseline zero-shot model initialization (and also re-training) techniques.

In Figure 7, we present the training curves of models trained during ablation as discussed in Section 4.2.

**Knowledge Distillation**    Recall we have already discussed distillation as a baseline in Section 4.1 and associated Figure 3. We perform an additional experiment in the same setting i.e. knowledge distillation as a baseline. Here we trained GPT-2 medium variants with 12 layers (half the number of a vanilla GPT-2 medium). We trained three models. First we distilled a 24-layer GPT-2 medium (teacher) to a 12-layer GPT-2 medium variant (student) and this student is initialized with all the alternate layers of the teacher. This setting is exactly same as discussed in DistillBERT Sanh et al. (2019). Next we trained two GPT-2 medium variants one from scratch (vanilla training) and the other with Inheritune recipe. Model trained with our recipe beats model trained with distillation. We defer a through investigation of distillation compared to Inheritune to future work.

**How Inheritune addresses Attention Degeneration?**    Recall we have discussed attention degeneration in Section 2 and attention patterns are visualized in Figure 2. Following up on our previous discussions in Figure 9 and Figure 10 we demonstrate that models trained with Inheritune has lesser *lazy layers* compared to it's larger counterpart trained form scratch. We performed rank analysis for Figure 9 utilizing vanilla 24-layer GPT-2 medium and our 16-layer GPT-2 variant trained using Inheritune. Additionally, we performed rank analysis for Figure 10 with a vanilla 48-layer GPT2 xLarge and a 24-layer GPT2 xLarge variant trained using Inheritune.

Recall we have previously discussed that attention degeneration is connected with vanishing gradients of keys and queries Noci et al. (2022). The vanishing gradients is caused when the norm of the gradients Bengio et al. (1994) are so small that it fails to generate meaningful back-propagation signal. Since we are training smaller models intuitively $\|W_Q\|$ and $\|W_K\|$ should be smaller compared to their larger counterparts and hence the norm of gradients in the case of smaller models derived using Inheritune is higher leading to better training.

## B    DEVELOPING A 1.5B SMALL BASE LM IN A LOW DATA REGIME WITH INHERITUNE

In this section, we aim to investigate the efficacy of Inheritune in a data and compute-constrained setting. We train a 1.5B parameter small base LM with only 1B tokens using a 3B parameter base LM on a single GPU (A6000) for less than half a day.

We assume the existence of a pre-trained reference model $\mathcal{M}_{\text{ref}}$, comprising $k$ layers, represented by $W_{\text{ref}} = \{W_0, W_1, \ldots, W_{k-1}\}$ trained with $\mathcal{D}_{\text{train}}$. However, this full training data is unavailable, and we only have a random tiny subset $\hat{\mathcal{D}}_{\text{train}} \sim \mathcal{D}_{\text{train}}$. We use OpenLLaMA-3B version 1 as the reference model pre-trained with 1T tokens from the RedPajama V1 dataset, which contains data

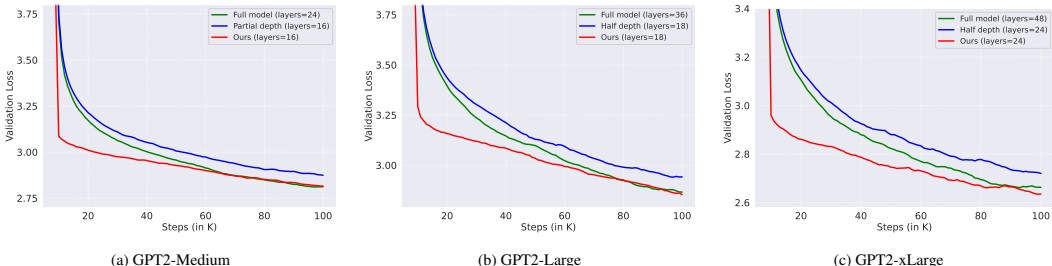

(a) GPT2-Medium          (b) GPT2-Large          (c) GPT2-xLarge

Figure 5: **Models derived using Inheritune converge faster and match the final validation loss of the full-sized model trained from scratch, despite being smaller.** Training GPT-2 xlarge, GPT-2 large and GPT-2 medium vanilla models from scratch and our variants with OpenWebText-9B for 100K steps.

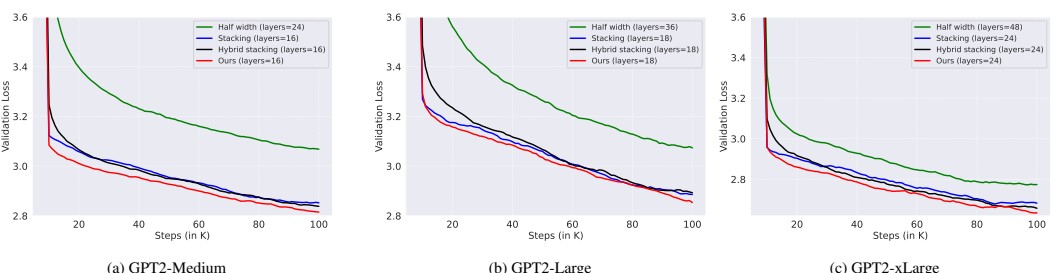

(a) GPT2-Medium          (b) GPT2-Large          (c) GPT2-xLarge

Figure 6: **Models derived using Inheritune outperform three zero-shot initialization and efficient training baselines in terms of final validation loss.** Our models demonstrate better convergence and generalization compared to all baselines. We trained GPT-2 xlarge, GPT-2 large and GPT-2 medium variants on OpenWebText-9B for 100K steps using baseline model initialization and efficient training techniques and our Inheritune training recipe.

from various domains such as common crawl, C4, Wikipedia, books, arXiv papers, GitHub, and Stack Exchange. We take 1B randomly sampled tokens[4] from the RedPajama dataset.

**Training recipe.** To adapt Inheritune for this new setting, we perform step 1 and step 2 in Algorithm 1 without growing the model (i.e., we skip step 3). We use the first $n = 13$ layers from our $k = 26$ layer reference model. We call our small base LM Ours-1.5B(#tokens). We train our model with data repetition for eight epochs (each epoch uses all the 1B tokens) with a batch size of 131K tokens per batch. We use 1 A6000 GPU for less than half a day of training. The choice of training epochs is based on the analysis provided later in this paper (refer to Figure 14). We use the lit-gpt framework for training all small base LMs discussed in this paper. Further discussions on the training hyper-parameters can be found in the next Section.

**Baseline models and evaluation.** We choose similarly sized (1-2B parameter) small base LMs trained with the RedPajama dataset and the reference base LM as primary baselines, as the quality of the pre-training data plays a key role in model development. We also include models OPT-1.3B Zhang et al. (2022) and Pythia-1.3B Biderman et al. (2023) as these models are pre-trained with a dataset similar to the RedPajama dataset. Table 6 lists the baseline models with their pre-training data.

In this study, we use few-shot accuracy, particularly 0-shot and 5-shot accuracy, on ten different downstream tasks to measure the quality of our 1.5B base LM. This evaluation of pre-trained base LLMs has been done in several prior works. Our evaluation methodology categorizes downstream tasks across four distinct genres: commonsense reasoning, natural language understanding, factuality, and natural language inference. We perform 0-shot evaluation for PIQA Bisk et al. (2020), BOOLQ Clark et al. (2019), WINOGRANDE Sakaguchi et al. (2020), WINOGRAD Kocijan et al. (2020),

---

[4]https://huggingface.co/datasets/togethercomputer/RedPajama-Data-1T-Sample

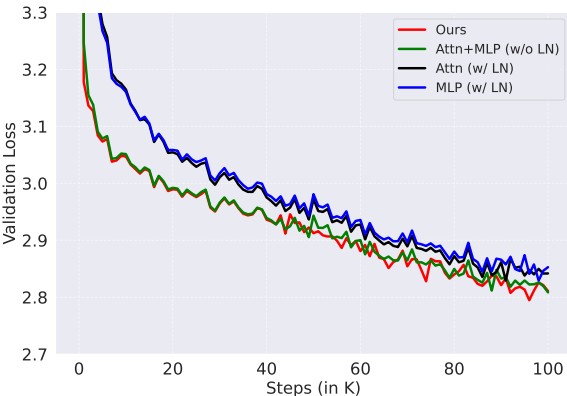

Figure 7: **Full training curves of 16-layer GPT-2 variants trained during ablations.** We analyze Inheritune approach while initializing some specific sub-modules in transformer blocks. Here, we initialize each transformer block of a 16-layer GPT-2 medium variant with three different configurations. First, we separately initialize attention and MLPs (FFNs) submodules; second, we initialize the attention and MLP weights while randomly initializing the layer norms. Finally, we perform Inheritune-initialize only the attention and MLP weights with all the respective layer norms.

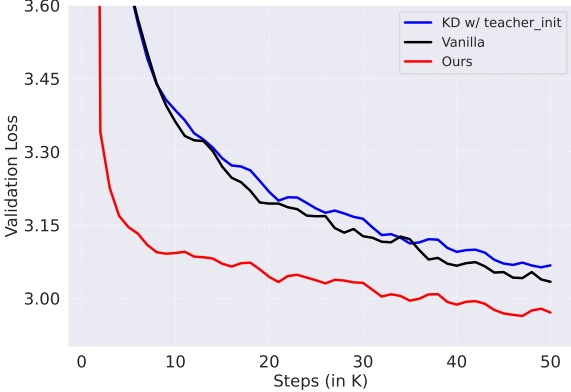

Figure 8: **A 12-Layer GPT-2 medium variant derived using Inheritune converges faster and generalizes better than a same-sized models trained from scratch and with Logit-based distillation with teacher initialization baseline.** Three 12-layer GPT-2 medium variants were trained: (1) a distilled model initialized with alternate layers from a 24-layer GPT-2 medium teacher, following the DistillBERT setup Sanh et al. (2019); (2) a model trained from scratch (vanilla training); and (3) a model trained using the Inheritune recipe. The model trained with Inheritune outperforms both the distillation-based model and the one trained from scratch, demonstrating the effectiveness of our approach.

LOGIQA Liu et al. (2020), TruthfulQA Lin et al. (2022), MNLI Bowman et al. (2015), QNLI Wang et al. (2018) and WNLI Wang et al. (2018) datasets. Next, we perform a 5-shot evaluation on the massive multitask language understanding benchmark (MMLU) Hendrycks et al. (2020). We use the lm eval harness framework Gao et al. (2024) for the entire evaluation.

## B.1 MAIN RESULTS IN LOW DATA REGIME

Table 5 presents a detailed performance evaluation across various tasks. Our 1.5B model, developed using Inheritune, excels in 7 out of 10 individual tasks. It achieves a score of 90% or higher compared to the reference language model, which is twice its size and trained with 1000 times more data, or it outperforms at least two other base LMs of similar size trained with 50-300 times more data. Favorable scores are highlighted in bold.

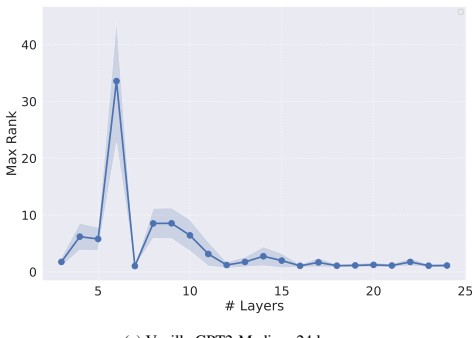
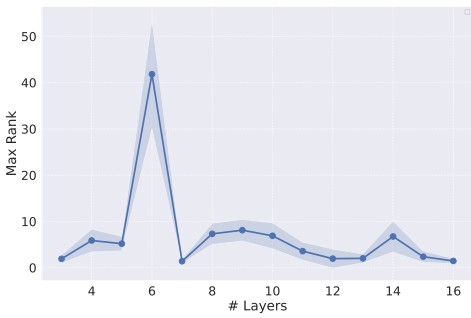

(a) Vanilla GPT2-Medium 24 layers

(b) Our GPT2-Medium 16 layer variant trained with Inheritune

Figure 9: **Rank collapse in deeper layers and its mitigation through** Inheritune**.** ==The maximum (max) rank across all attention heads for each layer is plotted, following the methodology in Fig. 1== (a) Analysis of a 24-layer GPT2 medium model reveals rank-1 attention matrices in later layers ==(those beyond the halfway point)==, indicating rank collapse. ==Specifically, 3 out of the last 12 later layers exhibit rank-1 attention matrices (mean rank accross all the 100 runs).== (b) Our 16-layer GPT2 medium variant, trained with Inheritune, demonstrates improved rank across all layers, highlighting the effectiveness of our approach. ==Notably, none of the later layers in our 16-layer variant exhibit rank-1 attention matrices.==

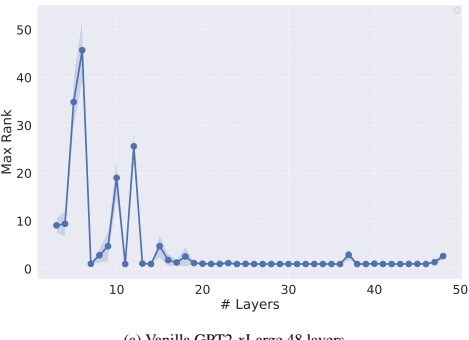
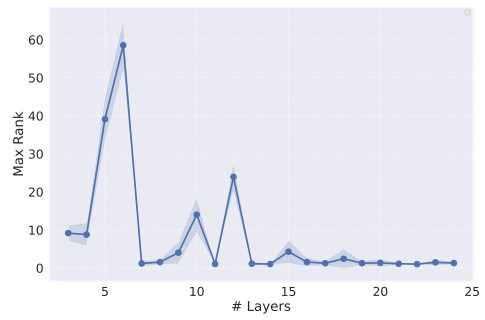

(a) Vanilla GPT2-xLarge 48 layers

(b) Our GPT2-xLarge 24 layer variant trained with Inheritune

Figure 10: **Rank collapse worsens for larger LLMs,** Inheritune **helps to mitigate rank collapse.** The maximum (max) rank across all attention heads for each layer is plotted, following the methodology in Fig. 1 (a) Analysis of a 48-layer GPT2 xLarge model reveals rank-1 attention matrices in later layers (those beyond the halfway point), indicating rank collapse. Specifically, 22 out of the last 24 later layers exhibit rank-1 attention matrices (mean rank across all the 100 runs). (b) Our 24-layer GPT2 xLarge variant, trained with Inheritune, demonstrates improved rank across all layers, highlighting the effectiveness of our approach. Notably, 2 out of 12 of the later layers in our 24-layer variant exhibit rank-1 attention matrices.

Next, we compare our small LM with the MPT-1.3B[5] model trained from scratch with 200B tokens of RedPajama dataset and find that we match 97% accuracy in all nine downstream tasks and the MMLU (5-shot) score. Additionally, we compare with OPT-1.3B and Pythia-1.3B models, showing that we outperform both in the MMLU (5-shot) score and perform comparably on the other nine datasets. This comparison illustrates that having a large reference base LM and a subset of its pre-training data allows the inherited target size base LM to be trained remarkably more sample-efficiently than training from scratch. Extended discussions on comparisons with the ShearedLLaMa model, generated by pruning and continual training from LLaMA2-7B, are provided in the supplementary materials.

**Ablation of Inheritune Across Different Model Sizes with 1B Tokens.** In the previous section, we considered a single choice of $n = k/2$, i.e., half the layers, for the size of the smaller model. Here, we investigate Inheritune with different choices of $n$, but the same 1B token dataset). All models use

---

[5] https://huggingface.co/mosaicml/mpt-1b-redpajama-200b

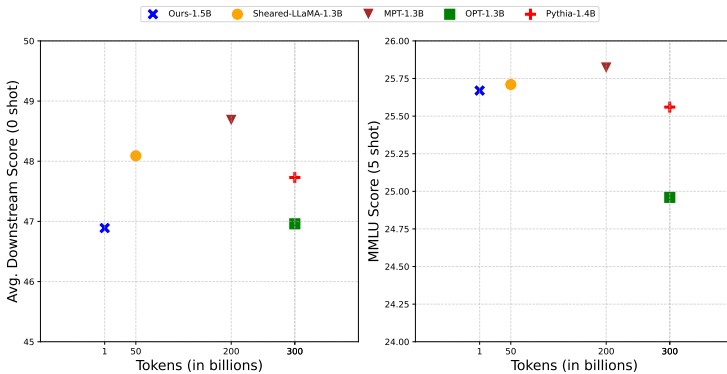

Figure 11: Performance of our 1.5B base LM derived using 1B data with Inheritune on an average of 9 different datasets (left) and MMLU benchmark (right) that evaluates commonsense, truthfulness, natural language inference and language understanding. We compare our model's performance with reference model-OpenLLamA-3B (2x size), other small base LMs of size 1B-2B parameters such as MPT-1.3B, OPT-1.3B, Pythia-1.4B (pre-trained from scratch) and ShearLLaMA-1.5B (pruned and continually trained using existing large base LM).

| Model | | Commonsense Reasoning | | | | |
|---|---|---|---|---|---|---|
| **Name** (# train tokens) | **Reference** | **Winograd** | **PIQA** | **Boolq** | **WinoGrande** | **Logiqa** |
| OpenLLaMA-3B (1T) | n/a | 63.46 | 74.97 | 67.18 | 62.27 | 28.4 |
| OPT-1.3B (300B) | n/a | 38.46 | 71.82 | 57.83 | 59.51 | 27.04 |
| Pythia-1.4B (300B) | n/a | 36.54 | 70.89 | 63.12 | 56.99 | 27.65 |
| MPT-1.3B (200B) | n/a | 63.46 | 71.44 | 50.89 | 58.09 | 28.26 |
| Sheared LLaMA-1.3B (50B) | LLaMA2-7B | 36.54 | 73.45 | 62.02 | 58.17 | 27.34 |
| **Ours-1.5B** (1B) | OpenLLaMA-3B | **50.96** | 56.47 | **61.68** | 51.69 | 25.19 |

| Model | | Lang. Understanding & Inference | | | | Factuality |
|---|---|---|---|---|---|---|
| **Name** ( # train tokens) | **Reference** | **MMLU(5)** | **WNLI** | **QNLI** | **MNLI** | **TruthfulQA** |
| OpenLLaMA-3B (1T) | n/a | 27.21 | 50.7 | 51.3 | 37.3 | 35 |
| OPT-1.3B (300B) | n/a | 24.96 | 42.25 | 51.29 | 35.82 | 38.67 |
| Pythia-1.4B (300B) | n/a | 25.56 | 53.52 | 49.48 | 32.76 | 38.66 |
| MPT-1.3B (200B) | n/a | 25.82 | 40.85 | 50.52 | 35.93 | 38.68 |
| Sheared LLaMA-1.3B (50B) | LLaMA2-7B | 25.71 | 49.3 | 50.98 | 37.94 | 37.14 |
| **Ours-1.5B** (1B) | OpenLLaMA-3B | **25.67** | **43.66** | **49.41** | **34.42** | **48.61** |

Table 5: **Our 1.5B model achieves performance comparable to baseline models despite being trained with fewer tokens.** Comparison of our target model ($\mathcal{M}_{\text{tgt}}$) derived using Inheritune with the reference model ($\mathcal{M}_{\text{ref}}$) and other baseline models of similar size when pre-trained from scratch and pre-trained with inherited weights and pruning. Although trained with fewer tokens, our model achieves performance comparable to the baseline models. We have highlighted all the scores in **bold** where our 1.5B model achieves at least 90% of the score compared to the reference LM or outperforms at least two of the baseline similar-size LMs. All the tasks are evaluated using 0-shot except MMLU, which is 5-shot. The models against which n/a is mentioned are trained from scratch.

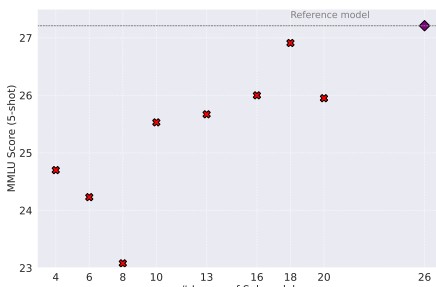

Figure 12: **Inheritune scales across multiple different model sizes.** Utilizing the OpenLLaMA-3B as a reference large base LM, demonstrates that multiple performant small base LMs of target size can be crafted using Inheritune with just 1B training tokens. The MMLU (5-shot) as a function of the number of submodels.

| Model | Training Data (# tokens) |
|---|---|
| OpenLLaMA-3B v1(ref) | RedPajama(1T) |
| Ours-1.5B[*] | RedPajama (1B) |
| Shear-LLaMA-1.3B[*] | RedPajama(50B) |
| MPT-1.3B | RedPajama(200B) |
| Pythia-1.4B | The Pile(300B) |
| OPT-1.3B | Custom data(300B) |

Table 6: **Comparison of training data across baseline models.** Overview of reference and baseline models, including their pre-training datasets and the number of tokens used during training. Note the significant variation in training data size, ranging from 1B to 1T tokens.

OpenLLAMA-3B as the large pre-trained reference model, with consistent training hyperparameters, changing only the choice of $n$.

We developed eight different submodels with $n = \{4, 6, 8, 10, 13, 16, 18, 20\}$. Figure 12 shows the MMLU (5-shot) score as a function of $n$. As expected, the trend line is positive-sloping. The submodel with 20 layers slightly decreases performance, potentially due to data overfitting as the model size increases. The training details for all these submodels are consistent with the target 1.5B small base LM and are detailed in the appendix. A more comprehensive investigation on the choice of $n$—including varying both $n$ and the number of training tokens jointly and evaluating a broader set of tasks is left for future work.

B.2   ADDITIONAL ANALYSIS WITH LARGER REFERENCE LMS AND 50B DATA

We further analyze Inheritune to see the impact of it's performance when more tokens are available. Initially for the main results we limited ourselves to 1B (i.e. 0.1%) tokens from the 1T pre-training data, here we use a 50B subset (i.e. 5%) of the pre-train data. Moreover we also extend this study to include larger base LMs of 7B parameters as reference models, employing OpenLLaMA-7B and LLaMA2-7B as reference models. For the purpose of this study we do not repeat the tokens from our 50B subset. As shown in Figure 13, we observe that there is clear improvement in overall MMLU (5-shot) score with more data. Additionally it is interesting to see that 1.5B (or 1.6B models) developed with Inheritune using larger reference models show even greater improvements when fed with 50B subset of non repetitive data (i.e fresh tokens). We present a Table 8 using Figure 13 to show the best MMLU (5-shot) scores achieved using different reference LMs. For developing our small base LMs using larger reference LMs we use $n$=7 (i.e. 7 layers). The training details are discussed in the following section.

**Ablations with number of epochs.**   We ran ablations (refer Figure 14) to choose the total number of epochs (multiple passes over the data) and observe that repetition when training our 1.5B (or 1.6B) LM is helpful particularly for MMLU. We also observe that the for an average of all the 9 other

| Models (# train tokens) | GPU Count | GPU Type | Time (# days) |
|---|---|---|---|
| MPT-1.3B (200B) | 440 | A100 | half |
| Pythia-1.4B (300B) | 64 | A100 | 4.6 |
| TinyLLaMA-1.1B (3T) | 16 | A100 | 90 |
| OPT-1.3B (300B) | 992 | A100 | – |
| Sheared LLaMA-1.3B (50B) | 16 | A100 | – |
| OpenLLaMA-3B (1T) | 256 | TPU v4 | 10 |
| **Our-1.5B (1B)** | **1** | **A6000** | **~half** |

Table 7: **Computational efficiency of Inheritune versus baseline models.** Comparison of pre-training compute requirements for publicly available small base LMs and our Inheritune-derived model. Metrics include GPU count, GPU type, and training duration, highlighting Inheritune's significant reduction in computational resources.

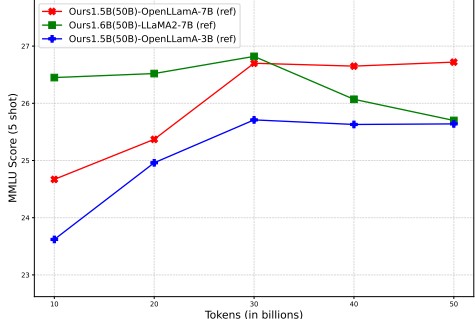

Figure 13: **Impact of reference model choice on Inheritune performance.** MMLU (5-shot) scores for 1.5B base LMs derived using Inheritune, trained on 50B unique tokens. Comparison across three reference models: OpenLLaMA-7B, LLaMA2-7B, and OpenLLaMA-3B. Results demonstrate Inheritune's effectiveness with various large language models as references.

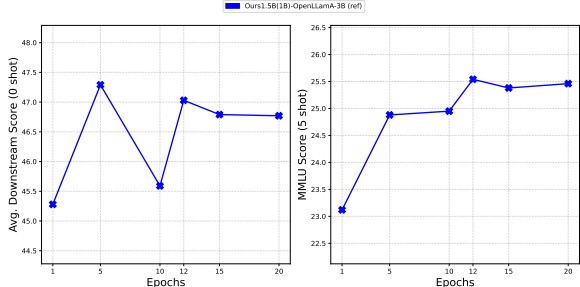

Figure 14: Performance of our 1.5B base LM derived using Inheritune based on existing OpenLLaMA-3B base model. Here we use 1B tokens and perform data repetition (epochs) during training. We further evaluate our model on an average of 9 different datasets (left) and MMLU benchmark (right).

| Model (# tokens), ref | MMLU(5) |
|---|---|
| Ours-1.6B (1B), LLaMA2-7B | 24.27 |
| Ours-1.5B (1B), OpenLLaMA-3B | 25.67 |
| Ours-1.5B (50B), OpenLLaMA-3B | 25.71 |
| Ours-1.6B (50B), LLaMA2-7B | 26.07 |
| **Ours-1.6B** (50B), OpenLLaMA-7B | **26.72** |

Table 8: Performance comparison of models on the MMLU (5-shot) task. Our models, even when trained with fewer tokens, show competitive performance compared to benchmarks. We have highlighted the best MMLU 5-shot score in **bold**.

| Model (# tokens) | Data type | MMLU (5-shot) |
|---|---|---|
| **Ours-1.5B** (1B) | 10 epochs | **24.95** |
| **Ours-1.5B** (50B) | 10B fresh | 23.62 |
| **Ours-1.5B** (1B) | 20 epochs | **25.46** |
| **Ours-1.5B** (50B) | 20B fresh | 24.96 |

Table 9: MMLU (5-shot) scores of Our-1.5B small base LM derived using 1B data for multiple data repetition–10 epochs and 20 epochs compared to the same model trained without data repetition for 10B and 20B fresh tokens. We derive all the variants of Our-1.5B small base using Inheritune with OpenLLaMA-3B as reference model. The models featured in this table correspond to those discussed in Figures 13 and 14.

datasets (i.e. except MMLU) peaks it's performance at 5 epochs and then deteriorates. Some prior works have studied this phenomenon that the scaling of downstream tasks with data is not always linear Biderman et al. (2023).

**To repeat or not to repeat the tokens.** Next we tackle the question – whether one should re-use 1B tokens for multiple epochs or use the same number of fresh tokens? Some prior works have recommended that if you have a reasonably large size dataset one can repeat it upto 4 epochs Muennighoff et al. (2023). In our study we observe that one can safely re-use 1B tokens upto 10-20 epochs as shown in Table 9. We emphasis that this phenomenon needs a through investigation in itself and we defer this to future work. The models discussed in Table are saved checkpoints during a single training run and not the final model unless otherwise specified.

### B.3 IMPLICATIONS OF LOW DATA REGIME

In this section, we discuss some of the key implications of our work in low data regime.

**Cheap and easy development of small base LMs.** Pre-training a small base LM of 1-2B parameters from scratch is extremely expensive. For instance mpt-1.3B base LM is pre-trained with 440 A100 GPUs for half a day, while the Pythia-1.4B base LM Biderman et al. (2023) utilized 64 A100-40GB GPUs for 4.6 days. Similarly, TinyLLaMA-1.1B model Peiyuan Zhang & Lu (2023) was pre-trained using 16 A100 GPUs for 3 months. Our 1.5B (1B data variant) LM shows competitive performance despite being trained with 1 A6000 GPU for less than 12 hours. The computational details are provided in Table 7, comparing the training resources of the baseline models listed in this paper. Typically small base LMs are finetuned for a specific task before deployment and are not used in it's base form. With Inheritune we present a really easy and cheap way for developing a small base LM to be later finetuned before deployment.

**Naive baseline for pre-training a scaled down variant of large base LMs.** Typically small variants of large base LMs are pre-trained using the same pre-training data Peiyuan Zhang & Lu (2023); Groeneveld et al. (2024). Our recipe introduces a new perspective of identifying sufficient

depth without losing any generalization on the held out validation set. Next, we also show that even with a small fraction of pre-train data (randomly sampled) and few initial layers of the large base LM one can develop a small base LM. Therefore our Inheritune recipe has the potential to become the naive baseline for any pre-training pipeline aiming to develop a smaller variant of a large base LM.

# C  IMPLEMENTATION DETAILS

## C.1  TRAINING DETAILS OF GPT-2 MODELS

For our main experiments, we focused on three sizes of GPT-2 models Radford et al. (2019): the vanilla GPT-2 xlarge with 1.5B parameters, GPT-2 large with 770M parameters and the vanilla GPT-2 medium with 355M parameters. We developed several variants of these models by adjusting the number of layers and hidden size. We trained all GPT-2 models with data repetition while using OpenWebText dataset, the trainset has 9B tokens and the validation set has 4.4M tokens. The key architectural configurations for the reference models, our models, and baseline models discussed in this paper are summarized in Table 10.

For all training runs, we used GELU activations, disabled bias terms, and removed dropout, following the nanoGPT codebase and Liu et al. (2023). We employed the AdamW optimizer with $\beta_1 = 0.90$ and $\beta_2 = 0.95$. The GPT-2 models were trained on a single node with 3 A100 GPUs (each with 40 GB of memory) using distributed data parallelism and gradient accumulation. In line with Liu et al. (2023), we scaled the attention logits inversely to the layer index across all GPT-2 models. Most hyperparameters were adapted from Liu et al. (2023), with key details provided below.

**Hyper-parameter details of GPT-2 Medium and variants.**

- Batch size: 50K tokens
- Learning rate: $3 \times 10^{-4}$,
- Warmup steps: 2K,
- Scheduler type: cosine decay to $\frac{1}{10}$ of max learning rate,
- Weight decay: 0.1,
- Gradient clipping value: 1,
- Total training steps: 100K

**Hyper-parameter details of GPT-2 large and variants.**

- Batch size: 16K tokens
- Learning rate: $2 \times 10^{-4}$,
- Warmup steps: 2K,
- Scheduler type: cosine decayed to $1 \times 10^{-5}$,
- Weight decay: 0.1,
- Gradient clipping value: 1,
- Total training steps: 100K

**Hyper-parameter details of GPT-2 xlarge and variants.**

- Batch size: 16K tokens
- Learning rate: $1.5 \times 10^{-4}$,
- Warmup steps: 2K,
- Scheduler type: cosine decayed to $1 \times 10^{-5}$,
- Weight decay: 0.1,
- Gradient clipping value: 1,
- Total training steps: 100K

**Hyper-parameter details of knowledge distillation training.**

We use the below loss for as our distillation based training loss. The validation loss is the student_loss.

$$\text{Total\_loss} = \alpha \cdot \text{student\_loss} + (1 - \alpha) \cdot \text{distillation\_loss}$$

- Model: 16-layer and 12-layer GPT-2 medium variants
- Softmax temperature: 1
- $\alpha$: 0.6
- Batch size: 50K tokens
- Learning rate: $3 \times 10^{-4}$,
- Warmup steps: 2K,
- Scheduler type: cosine decay to $\frac{1}{10}$ of max learning rate,
- Weight decay: 0.1,
- Gradient clipping value: 1,
- Total training steps: 50K

| Models | Layers | Hidden Size | Heads | Variant | |
|---|---|---|---|---|---|
| GPT2-xlarge(1.5B) | 48 | 1600 | 25 | Original | |
| GPT2-large(770M) | 36 | 1280 | 20 | Original | **Reference models** |
| GPT2-large$^{\dagger}$(680M) | 32 | 1280 | 20 | Original | |
| GPT2-medium(355M) | 24 | 1024 | 16 | Original | |
| GPT2-large | 18 | 640 | 10 | half width | **Init. baselines** |
| GPT2-medium | 16 | 512 | 8 | half width | |
| GPT2-xlarge | 24 | 1600 | 25 | Ours | |
| GPT2-large | 18 | 1280 | 20 | Ours | **Our variants** |
| GPT2-large$^{\dagger}$ | 16 | 1280 | 20 | Ours | |
| GPT2-medium | 16 | 1024 | 16 | Ours | |

Table 10: Overview of all the GPT2 models used in this study and their architectural configurations. The model configurations of stacking and hybrid stacking are same as our variants.

## C.2 TRAINING DETAILS OF 1.5B OPENLLAMA MODEL

**Small base LMs trained with 1B data**    We present our main results with Our-1.5B model trained with an existing OpenLLaMA version 1 Geng & Liu (2023) and 1 B tokens randomly sampled from 1T redpajama version1 data. The hyper-parameters related to this model is provided below. It is important to note that our claim that we only use 1 GPU for less than 12 hours to train Our-1.5 B model is specific to models derived using Inheritune with 1B data. Next we also train multiple sub-models as shown in Figure 12 the training details remains consistent with that of the initial model discussed earlier. However we observe that increasing the number of layers in a sub-model also increase the training time.

**Hyper-parameter details of our 1.5B base LM derived using OpenLLaMA-3B as refernce LM:**

- Training tokens: 1B
- Training epochs: 8
- Training steps: 64K
- Learning rate: $3 \times 10^{-4}$
- Scheduler: Cosine
- Weight decay: 0.1
- Optimizer: AdamW

- Warm up steps: 1000

- Batch size: 131K

- GPU count: 1

- GPU type: A6000

- GPU hours: $\sim 8$ hours

- GPU hours/epoch: $\sim 54$ minutes

**Training details of small base LMs with 50B data.**  We also trained our 1.5B model with larger subsets of data as shown in Figure 13.It is important to note that all the intermediate tokens until 50B are intermediate checkpoints of a single training run. Some of the key hyper-parameters of our training runs are discussed below. We have also trained three variants of small base LMs utilizing 3 different reference base LMs namely OpenLLaMA-3B, OpenLLaMA-7B and LLaMA2-7B. For target LMs developed with OpenLLaMA-3B we use $n$=13 i.e. 13 layers. For target LMs developed using reference LMs of 7B parameters we use $n$=7 i.e. 7 layers. The training hyper-parameters remains consistent across all the models trained with 50B subset of the pre-train data.

**Training hyper-parameters of our target 1.5B and 1.6B small base LMs:**

- Training tokens: 50B

- Training epochs: $\sim 1$

- Training steps: 191K

- Learning rate: $3 \times 10^{-4}$

- Scheduler: Cosine

- Weight decay: 0.1

- Optimizer: AdamW

- Warm-up steps: 1000

- Batch size: 131K tokens

- GPU count: 1

- GPU type: A100

- GPU hours: $\sim 18$ hours

## D    EXTENDED DISCUSSION

### D.1    DISCUSSION ABOUT ATTENTION SINK

The term "attention sink" Xiao et al. (2024) refers to the phenomenon where the first token in a sequence receives disproportionately high attention scores compared to other tokens in the attention maps. While there is some connection with Inheritune, as we have also observed that many attention matrices are not only rank-1 but also single-column (with all attention scores concentrated on the first token), this connection has not been explicitly established in Xiao et al. (2024) with respect to rank-1 behavior or poor training of later layers.

In contrast, as illustrated in Figure 1, we compute the maximum rank of all attention matrices within a layer. For instance, consider a layer where only 2 out of 5 attention heads exhibit attention sink behavior. This does not make the layer lazy, as attention is computed as a concatenation of activations across all heads. A lazy layer, however, has all 5 out of 5 attention heads fully degenerated, with their attention matrices being rank-1. We provide evidence that such lazy layers are indicative of poorly trained layers.

| Layers | Initialization | Avg max ranks | Val Loss (↓) |
|---|---|---|---|
| 4 | rand | n/a | 3.25 |
| 4 | 1-4 layers from vanilla GPT2 | 8.40 | 3.22 |
| 4 | 5-8 layers from vanilla GPT2 | 9.48 | 3.19 |
| 4 | 9-12 layers (lazy layers) from GPT2 | 1.22 | 3.23 |

Table 11: **Impact of initialization strategies on GPT2-small variants.** We analyzed the rank characteristics of a vanilla GPT2-small model (125M, 12 layers) trained on OpenWebText for 100K steps. Four-layer GPT2-small variants were initialized using the first 4 layers [1–4], middle 4 layers [5–8], last 4 layers [9–12], or with random initialization, and then trained for 100K steps on OpenWebText. Models initialized with the last 4 layers performed similarly to random initialization, while those initialized with layers exhibiting higher average max ranks achieved the best validation loss, regardless of proximity to the embedding layer. The training plots and rank analysis are provided in Figure 15.

## D.2 FURTHER INVESTIGATION ON LAYERWISE MODEL INITIALIZATION

One may argue that initialization with initial layers (early layers before the halfway point) works best cause they are closer to the embedding compared to the later layers (those beyond the halfway point). This may not be the right interpretation and below we provide evidence that layerwise max rank of attention matrices (as discussed in Figure 1) provides stronger signal for selecting layers.

We trained a vanilla GPT2-small (125M) model with 12 layers for 100K steps using the OpenWebText dataset. First, we conducted a rank analysis of this model, as shown in Figure 15. Next, we trained three GPT2-small variants for 100K steps, each with four layers initialized from the vanilla GPT2-small model: (a) the first four layers [1, 2, 3, 4], (b) the middle four layers [5, 6, 7, 8], and (c) the last four layers [9, 10, 11, 12]. In addition, we trained another GPT2-small variant with random initialization for 100K steps all using OpenWebText. The key results are presented in Table 11, and the complete training plots are shown in Figure 15. In summary, we observed that initializing the model with layers closer to the embedding did not yield the best final validation loss (lower is better). Instead, model initialized with layers from the vanilla GPT2-small model with average higher max ranks (as indicated by Avg Max Ranks in Table 11) demonstrated the best performance.

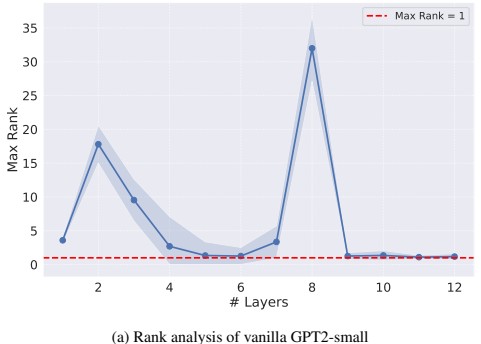

(a) Rank analysis of vanilla GPT2-small

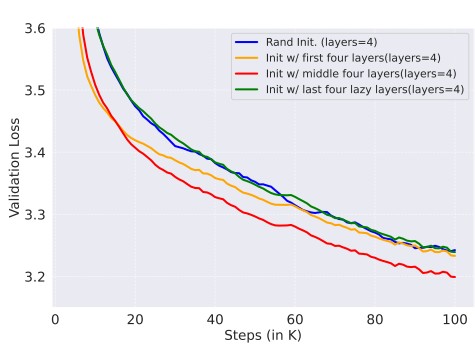

(b) GPT2-small 4-layer variants trained with various initializations.

Figure 15: **Early layers from reference models used in** Inheritune **for target model initialization perform best due to their higher average max ranks, not their proximity to the embedding layer.** a) Rank analysis of a vanilla GPT2 small model (125M) with 12 layers trained with OpenWebText for 100K steps. b) We initialize 4-layer GPT2-small variants with first 4 layers [1–4], middle 4 layers [5–8], last 4 layers [9–12], and with random initialization. We trained thses models for 100K steps using OpenWebText. Models initialized with last 4 layers performs close to random. Models with layers showing higher average max ranks achieved the best validation loss, not those closer to the embedding. Please also refer Table 11).

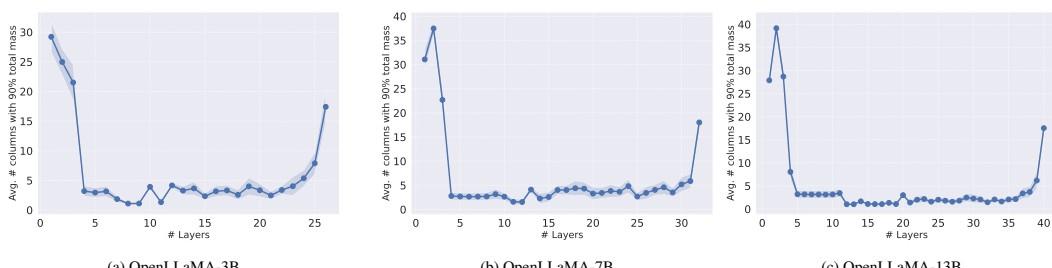

(a) OpenLLaMA-3B        (b) OpenLLaMA-7B        (c) OpenLLaMA-13B

Figure 16: **The overall mass of attention matrices in billion-scale LLMs, pre-trained on trillions of tokens, tends to concentrate in fewer columns. This phenomenon becomes increasingly pronounced as the model size grows.** We computed attention matrices using 100 tokens from a random subset of RedPajama with 1B tokens. Next, we performed 100 runs and plotted the mean and standard deviation of the mass as a function of layers for our mass analysis, respectively. We followed the same procedure as discussed in Section 2. Pre-trained checkpoints of OpenLLaMA-3B, OpenLLaMA-7B, and OpenLLaMA-13B (Geng & Liu, 2023), trained on 1T tokens from the RedPajama dataset Computer, 2023, were utilized. Overall, we observed that 90 of the total mass of the attention matrices resides in fewer columns, with many attention matrices in the OpenLLaMA-13B model being single-column. This observation aligns closely with our analysis in Figure 1.

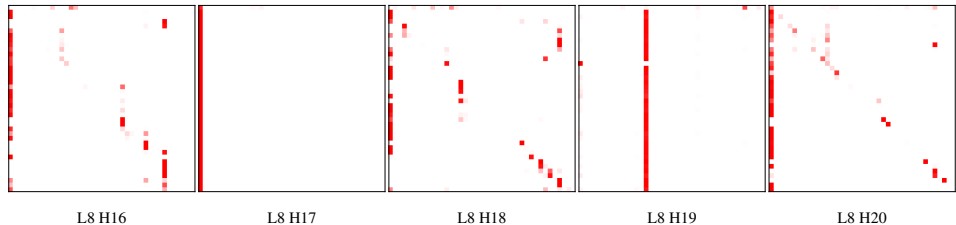

A **non-lazy layer** of a pre-trained GPT2 xLarge 48 layer model.

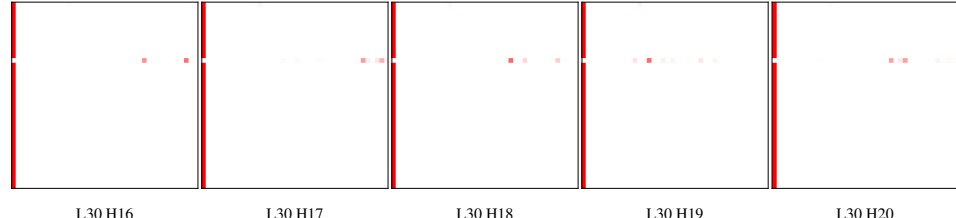

A **lazy layer** of a pre-trained GPT2 xLarge 48 layer model.

Figure 17: **Visualization of attention patterns in lazy and non-lazy layers of a vanilla GPT-2 xLarge model with 48 layers.** The top row displays attention patterns for various heads (H) in layer (L) 8, while the bottom row shows patterns for layer (L) 30. We observe attention sinks (Xiao et al., 2024) in nearly all attention patterns across both lazy and non-lazy layers.