# OpenReview forum: "Inheritune: Training Smaller Yet More Attentive Language Models"
_ICLR.cc/2025/Conference — Submitted to ICLR 2025_

### Official Review · Reviewer_FevH · 2024-11-03

**Soundness:** 2
**Presentation:** 3
**Contribution:** 2
**Rating:** 5
**Confidence:** 3

**Summary:**

This paper highlights attention degeneration, where layers become rank-1 matrices, leading to ineffective learning in "lazy layers." The proposed method initializes a smaller model with a few layers from a larger pre-trained model and progressively adds layers, enabling efficient training of high-performing models. Experiments across various GPT-2 sizes show that Inheritune maintains or improves performance compared to larger models trained from scratch or with other initialization methods, establishing it as an effective strategy for compact language models.

**Strengths:**

The paper is well-written and easy to understand, with a logical flow that supports the arguments presented. Additionally, it demonstrates the superiority of the proposed method through a variety of experiments.

**Weaknesses:**

It seems a bit too intuitive to refer to this as an initialization method utilizing the lazy layer phenomenon, even though this paper has identified the occurrence of lazy layers. The logic of using early layers for initialization because of lazy layers feels somewhat weak. When I consider that there are 36 teacher layers and I need to initialize 18 layers, it appears quite obvious to select the first 18 layers, as they are likely to be more compatible with the embeddings.

The experiments presented in the paper are largely conducted with a limited number of steps, which raises questions about the actual performance. It seems that the approach does not significantly differ from simply trimming a larger model and fine-tuning it, leading me to wonder whether the results truly reflect superior performance.

**Questions:**

(1) I think the phenomenon of lazy layers is similar to the attention sink phenomenon. Does approaching rank 1 mean that attention is concentrated on a specific token?

(2) In Figures 1 (c) and (f), rather than suggesting that the lazy layers are unnecessary, I suspect that the early layers are more compatible with the embeddings. Could you perhaps compare using the first nine layers with using the remaining ones randomly?

(3) This paper adopts a three-step approach of Inherit, Train, and Grow. What are the advantages of this method compared to reaching the final size all at once?

(4) I would like to inquire about the comparison between the GPT-2 Medium 16-layer variant trained from scratch and the one trained with Inheritune in Figure 2, 9. Does Inheritune effectively prevent rank collapse?

---

> ### Author Response · Authors · 2024-11-24
> **Author's Rebuttal 1/2**
>
> Thank you for taking time to review our work and providing feedback. We have addressed all your concerns here and updated the paper (updated text marked in yellow) accordingly.
>
> ### **Argument about Initialization with Initial layers works as they are closer to embedding**
>
> **Clarification:** We would like to highlight that the rank-1 ness (attention degeneration phenomenon) of attention matrices as a proxy to understand how well layers of a specific model are trained is novel and hasn't been reported in any prior works. Moreover this results are not intuitive as well.
>
> **Additional Experiments and Results:** As you have argued that initialization with initial layers works best cause they are closer to the embedding compared to the later layers (those beyond the halfway point). This may not be the right interpretation and below we provide evidence that layerwise max rank of attention matrices (as discussed in Fig.1 of original paper) provides stronger signal.
>
> We trained a vanilla GPT2-small (125M) model with 12 layers for 100K steps using the OpenWebText (OWT-9B) dataset. First, we conducted a rank analysis of this model, as shown in New Figure 1 (https://postimg.cc/NKcH5rfX).
>
> Next, we trained three GPT2-small variants for 100K steps, each with four layers initialized from the vanilla GPT2-small model: (a) the first four layers [1, 2, 3, 4], (b) the middle four layers [5, 6, 7, 8], and (c) the last four layers [9, 10, 11, 12]. In addition, we trained another GPT2-small variant with random initialization for 100K steps all using OWT-9B. The key results are presented in New Table 1, and the complete training plots are shown in New Figure 2 (https://postimg.cc/svSWjJyy). We have also updated the paper draft to reflect these changes. In summary, we observed that initializing the model with layers closer to the embedding did not yield the best final validation loss (lower is better). Instead, model initialized with layers from the vanilla GPT2 that exhibited higher max ranks (as indicated by Avg Max Ranks in New Table 1) demonstrated the best performance.
>
> | Layers | Initialization                                   | Avg max ranks |  Val Loss ↓|
> |--------|----------------------------------------|--------------|----------|
> | 4      | rand                                   | n/a          | 3.25     |
> | 4      | 1-4 layers from vanilla GPT2          | 8.40         | 3.22     |
> | 4      | 5-8 layers from vanilla GPT2          | 9.48         | 3.19     |
> | 4      | 9-12 layers (lazy layers) from GPT2   | 1.22         | 3.23     |
>
> **New Table 1: 4 layer GPT2-small variant initialized with layers with higher max ranks achieves superior val loss (lower is better). Ablation of 4 layer GPT2-small model variants initialized with first four, middle four and last four layers (these are lazy layers) of vanilla GPT2-small model with 12 layers. All models are trained for 100K steps with OWT-9B data.
>
> >The experiments presented in the paper are largely conducted with a limited number of steps, which raises questions about the actual performance.
>
> We have inherited the pre-training set up from [2]v1. For pre-training experiments in many prior works such as [2],[3],[4], and [5] GPT2 models are trained for 100K steps using OpenWebText (OWT-9B). Moreover in one of our baseline in Table 1 (in the original paper) we have trained our baseline GPT2 variants from scratch for 200K steps. We emphasis that our recipe works well on GPT2-medium, GPT2-large and GPT2-xLarge variants and this provides a stronger signal of the efficacy of our method.
>
> References
>
> [1] Xiao et al., Efficient Streaming Language Models with Attention Sinks.
>
> [2] Liu et al., Sophia: A Scalable Stochastic Second-order Optimizer for Language Model Pre-training.
>
> [3] Sanyal et al., Early Weight Averaging meets High Learning Rates for LLM Pre-training.
>
> [4] Taniguchi et al., ADOPT: Modified AdamCanConverge with Any β2 with the Optimal Rate.
>
> [5] Fu et al., Hungry Hungry Hippos: Towards Language Modeling with State Space Models.

---

> > ### Author Response · Authors · 2024-11-24
> > **Author's Rebuttal 2/2**
> >
> > > Q1) I think the phenomenon of lazy layers is similar ...
> >
> > **Attention Sinks Does Not Necessarily imply Lazy Layers**
> >
> > The term "attention sink" [1] refers to the phenomenon where the first token in a sequence receives disproportionately high attention scores compared to other tokens in the attention maps. While there is some connection, as we have also observed that many attention matrices are not only rank-1 but also single-column (with all attention scores concentrated on the first token), this connection has not been explicitly established in [1] with respect to rank-1 behavior or poor training of later layers.
> >
> > In contrast, as illustrated in Figure 1 of our paper, we compute the **maximum rank** (plotted on the y-axis) of all attention matrices within a layer. We define a layer as "lazy" only if the maximum rank of its attention matrices is 1, as discussed in Section 2. For instance, consider a layer where only 2 out of 5 attention heads exhibit attention sink behavior. This does not make the layer lazy, as attention is computed as a concatenation of activations across all heads. A truly lazy layer, however, has all 5 out of 5 attention heads fully degenerated, with their attention matrices being rank-1. We provide evidence that such lazy layers are indicative of poorly trained layers.
> >
> > > Q2 In Figures 1 (c) and (f), rather than suggesting that the lazy layers are unnecessary ...
> >
> > We didn't claim that lazy layers are unnecessary as we utilize many of the lazy layers while growing (step 4 of algorithm 1) the model during Inheritune training. Refer previous discussion about proximity to embedding.
> >
> > >Q3 This paper adopts a three-step approach of Inherit, Train, and Grow ...
> >
> > Sure if the final size where the smaller model has the same val loss is known then one can start those many number of layers, but oftentimes it will be not know. As shown in Table 1 from original paper we trained multiple variants of GPT2 medium model to get to the final 16 layer varaint that beats the vanilla 24 layer variant.
> >
> > >Q4  I would like to inquire about ...
> >
> > Yes, Inheritune mitigates attention degeneration (rank collapse in lazy layers) in both GPT2-medium variant and GPT2-xLarge variant. Refer Fig. 9, Fig. 10 and related discussion supplementary sec A in the updated version. I have also posted Fig. 9 (https://postimg.cc/bGKwCcXc) and Fig. 10 (https://postimg.cc/cKKJB1bB).
> >
> > Overall, we observe that attention degeneration worsens for larger LLMs; Models trained with Inheritune observes lesser attention degeneration.

---

> > > ### Author Response · Authors · 2024-11-28
> > > **Request to respond by Authors: Addressed concerns with additional experiments**
> > >
> > > Dear Reviewer FevH,
> > >
> > > Following up after previous discussion. You have suggested an alternate hypothesis on why this method works, and we have ran additional experiments provided evidence on efficacy of our method and also discussed why closer to embedding interpretation may not hold in language models. We would be very grateful if you may respond to our rebuttal.

---

> > > > ### Comment · Reviewer_FevH · 2024-11-28
> > > >
> > > > The idea of identifying lazy layers and utilizing them is promising. However, the proposed method appears somewhat naive as it heavily relies on experimental results. While the effectiveness of the approach has been demonstrated through massive experiments, when considering whether we would actually utilize this, a few additional questions and concerns arise.
> > > >
> > > > 1) whether the perplexity (ppl) improvements in Table 1 and Table 2 are significant enough,
> > > > 2) why Table 2 focuses solely on perplexity without comparing other metrics?
> > > > 3) whether lazy layers are consistently identified across all datasets or vary depending on the data.
> > > >
> > > > The author's thorough responses address most of these initial questions, so I decided to raise my score to 5 points.
> > > > As a Language Model researcher, I believe it would have been better if the observations in this study had been utilized more effectively.

---

> ### Author Response · Authors · 2024-12-01
> **Response to Additional Questions: Thanks for increasing the score and calling this work promising**
>
> Thank you for improving the score and indulging in further discussions.
>
> > However, the proposed method appears somewhat naive as it heavily relies on experimental results.
>
> Prior influential works such as LoRA, QLoRA etc started as an empirical paper. Just because an observation is simple and is studied only through experiments does not necessarily mean the idea is weak.
>
> > Q1 whether the perplexity (ppl) improvements in Table 1 and Table 2 are significant enough.
>
> A small improvement in training/validation perplexity during pre-training can be highly significant. For example, consider the LLaMA (version 1) models [6]. The LLaMA-7B model achieves a loss (log perplexity) of 1.8, while the LLaMA-13B model achieves a slightly lower loss of 1.74, with both models trained on 1 trillion tokens. See image taken from [6] (https://postimg.cc/0r41KtkV). See the full training curves of Table 1 and Table 2 in supl. Fig 5 and 6. Models trained with Inheritune also outperforms models trained from scratch in downstream performance.
>
> > W2 why Table 2 focuses solely on perplexity without comparing other metrics?
>
> Because the pre-train loss is known to be well correlated with the zero-shot performance both theoretically [7] and empirically [6].
> Moreover in Table 2 all the models are **not trained from scratch** hence we established an apples-to-apples comparison using val loss.
>
> >W3 whether lazy layers are consistently identified across all datasets or vary depending on the data.
>
> **Yes, lazy layers can be consistently identified across many datasets and also billion size LLMs (trained with 1T tokens)**
>
> Great feedback, we performed additional experiments. In Fig 1 we have utilized vanilla GPT2 models trained with OpenwebText (OWT) and analysis was conducted using val set of OWT (refer Section 2). Next we took the same models trained with OWT but now we perform rank and mass analysis using **new data Fineweb_edu**( process remains the same). In New Figure (https://postimg.cc/4nPdR3yc) one can clearly see lazy layers in vanilla GPT2 medium , large and Xlarge models. Our analysis is quite robust. We will update the draft accordingly.
>
> **Mass Analysis of OpenLLaMA 3B, 7B and 13B trained with 1T tokens.***
>
> We conducted mass analysis (same way as discussed in Sec 2) for OpenLLaMA 3B (26 layers), 7B (32 layers) and 13B (40 layers) models. Refer Fig 16 in updated draft. Posting the Figure here as well (https://postimg.cc/0bbpy2Q6). We observe that 90% of the total mass of the attention matrices are concentrated in fewer columns i.e. in less than 5% of the total columns (i.e 100 for a 100x100 attention matrix) for 3B and 7B models. Next many of columns in later layers are single column for the 13B model.
>
> Next with Openllama 3B as reference model we developed a target model of 1.5B while only using 0.1% of the original pre-training tokens (only 1B tokens) whereas the reference model was trained using 1000B (1T) tokens. Refer supplementary Section B.
>
> Again thank you for your helpful feedback. We have worked very hard to answer all your concerns, we would be very grateful if you may re-consider your score.
>
> **References**
>
> [6] Touvron et al., LLaMA:Open and Efficient Foundation Language Models.
>
> [7] Nikunj Saunshiet al., A mathematical exploration of why language models help solve downstream tasks.

---

> > ### Author Response · Authors · 2024-12-02
> > **Request to Respond by Authors**
> >
> > Dear Reviewer FevH,
> >
> > Thank you for your thoughtful remarks. We have added additional experiments to address your remark on varying data for analysis. Meanwhile we if you believe we have addressed your concerns well enough please consider improving your scores.

---

### Official Review · Reviewer_Bigz · 2024-11-04

**Soundness:** 2
**Presentation:** 3
**Contribution:** 2
**Rating:** 6
**Confidence:** 4

**Summary:**

This paper studies and proposes a better way to create smaller LLMs from existing pretrained models, in particular by examining whether the learned attention layers exhibit meaningful attention patterns or degenerate into single-column structures (“lazy layers”). They find later layers are more lazy across GPT-2 LLMs, and propose Inheritune as a method to exploit this phenomenon. Inheritune works by initializing a smaller model with the first half of the layers from a pre-trained larger LLM, training this model, and then progressively growing it by adding more blocks if necessary. This process is repeated until the smaller model's performance matches or surpasses that of the reference model. Through experiments using GPT-2 models of varying sizes, on datasets like OpenWebText-9B and FineWeb_edu, and using evaluations including language modeling perplexity and downstream LM Evaluation harness tasks, the authors demonstrate that Inheritune consistently outperforms various baselines, including larger models trained from scratch, models initialized with different techniques (stacking, hybrid stacking, half-width), and models trained using knowledge distillation. They further provide interesting analysis into Inheritune's behavior via various ablations on target modules and pretraining data mix.

**Strengths:**

**Interesting study + method contribution**
I liked the combination of studying a natural phenomenon (lazy layers showing up in pretrained decoder LLMs), and further exploiting this to create a well-motivated method for efficiently creating smaller models.

**Comprehensive Task Evaluation**

I appreciated how the authors evaluated not just validation-set perplexity, but also zero-shot performance on WikiText and Lambada in main experiments, and further considered popular LM Evaluation Harness tasks.

**Good Performance**

The experiments demonstrate Inheritune's strong performance compared to baselines such as randomly initializing models, stacking, hybrid stacking, and knowledge distillation.


**Interesting Ablation Studies for In-depth Analysis**

I appreciated the study into the different aspects of Inheritune, including initializing different submodules, using different pretraining data mixes.

**Weaknesses:**

**Insufficient treatment of related work / first contribution novelty**

How does the notion of "lazy layers" relate to prior work such as Attention Sinks [1]? In particular, the claim that “Notably, this phenomenon has not been studied in the context of standard LLMs.” (L043) does not seem true? See Attention Sinks [1] at ICLR 2024, which studied and observed before that several standard decoder-only LLMs (Llama 2 7B, Pythia 12B, Falcon 7B, MPT-7B) display this “lazy layer” phenomenon.

The method and way to exploit this lazy layer phenomenon seem novel, but I do think there needs to be better discussion on how the lazy layer contribution is novel.

**Limited model diversity (model family and scale)**

The study and method are limited to a single model family (GPT-2), where the largest LLM evaluated is only 1.5B parameters. Overall, the paper would be stronger if the findings were shown beyond a single class of models, and on a variety of more modern and popular models (e.g., Llama 3 8B, Mistral 7B, Phi 1.5, Gemma, etc.). While I don't fault the authors for not evaluating 7B models due to budget constraints  (although doing so would increase my score); even under the 1.5B parameter budget, we have models available such as Phi 1.5 from 2023 [2].

**Motivation behind comparisons**
I found some of the comparisons a bit too "baseline", but this could be clarified with discussion on their motivation.
* For example, if the motivation of Inheritune is to save model memory, how does the method compare to quantization techniques that can drastically reduce the parameter memory?
* If using less layers can improve inference or generation efficiency (as a benefit over quantization techniques), it would be good to see this benchmarking analysis


[1] Efficient Streaming Language Models with Attention Sinks, https://arxiv.org/abs/2309.17453

[2] Textbooks Are All You Need II: phi-1.5 technical report https://arxiv.org/abs/2309.05463


**--- After rebuttal revisions ---**
I think the authors did a reasonable job clarifying the distinction between lazy layers and attention sinks, while also extending their analysis to pretrained LLMs outside of GPT-2 architectures trained on OWT.

I still think the paper's presentation + method could be improved, e.g., explicitly using the number of observed  lazy layers to recommend how many layers should be kept (instead, they keep this main design choice for pruning at `num_layers // 2`, which makes the lazy layer connection a bit vacuous imo).

For this I am willing to raise my score to 6.

**Questions:**

Why are the target / smaller models initialized as k/2, if k is the number of layers in the original model? This seems a bit ad-hoc. Was there any study into if lazy layers were much more frequent in the second-half of the LLMs? Plotting or visualizing this would be support this design choice more.
* Because you have the LLMs, some analysis into whether a layer is lazy or not, and using this to inform which layers should be preserved, might also improve the method quality.

Is the step-count comparison fair? To make an Inheritune model, we need to have a full pretrained model in the first place. Are the steps it takes to acquire this pretrained model factored into the total step comparisons?
* Conversely, when making the claim that Inheritune outperforms randomly initialized models with the original parameter counts, are these full models further trained to hit the same total training updates that it takes to create Inheritune models?

In Figure 2, if the GPT2 models are decoder-only / causal, should the attention weights only be limited to lower-triangular?

---

> ### Author Response · Authors · 2024-11-24
> **Rebuttal by Authors 1/2**
>
> Thank you very much for the thorough review and helpful feedback. You have raised some very interesting questions which we have addressed below.
>
> >W1 Insufficient treatment of related work / first contribution novelty
>
> **Attention Sinks Does Not Necessarily imply Lazy Layers**
>
> The term "attention sink" [1] refers to the phenomenon where the first token in a sequence receives disproportionately high attention scores compared to other tokens in the attention maps. While there is some connection, as we have also observed that many attention matrices are not only rank-1 but also single-column (with all attention scores concentrated on the first token), this connection has not been explicitly established in [1] with respect to rank-1 behavior or poor training of later layers.
>
> In contrast, as illustrated in Figure 1 of our paper, we compute the **maximum rank** (plotted on the y-axis) of all attention matrices within a layer. We define a layer as "lazy" only if the maximum rank of its attention matrices is 1, as discussed in Section 2. For instance, consider a layer where only 2 out of 5 attention heads exhibit attention sink behavior. This does not make the layer lazy, as attention is computed as a concatenation of activations across all heads. A truly lazy layer, however, has all 5 out of 5 attention heads fully degenerated, with their attention matrices being rank-1. We provide evidence that such lazy layers are indicative of poorly trained layers. We have cited [1] in our updated draft and added some discussion about it.
>
> We have cited [1] in our updated draft and added some discussion about it. We emphasis that the lazy layer phenomenon has some distinction from attention sinks and hence we claimed this as novel.
>
> > W2 Limited model diversity (model family and scale)
>
> This is actually a great feedback. The attention maps are very sensitive to data distribution. Since we have trained the models with OpenWebText (OWT-9B) and then used the validation set to plot the attention maps we made sure that there is little to no distribution shift. This is very important for our setting because we are using rank-1 ness (attention degeneration) as a proxy for evaluating how well the layers are trained.
>
> For the Gemma and Phi models, neither the training nor validation datasets are publicly available. Even if they were, we lack the computational resources required to train inheritune variants of these models, as they are trained on trillions of tokens.
>
> > Motivation behind comparisons I found some of the comparisons ...
>
> The primary motivation of Inheritune is to leverage the rank of attention matrices as a proxy for assessing how well the layers are trained. In decoder-style deep LLMs, gradient vanishing often occurs [2,3], leading to undertrained layers, particularly in the later stages. For much larger (and deeper) models with poorly trained layers, Inheritune offers a solution by enabling the model to become smaller while addressing attention degeneration.
>
> Smaller models typically translates to lower memory requirements and faster generation. In this paper, we primarily focused on validation loss (generalization) and downstream performance following prior works [4,5]. The key contribution lies in demonstrating that a smaller model can achieve the same validation loss as a much larger model, highlighting its efficiency without compromising performance.
>
> ---
> References
>
> [1] Xiao et al., *Efficient Streaming Language Models with Attention Sinks.*
>
> [2] Barbero et al., *Transformers need glasses! information over-squashing in language tasks.*
>
> [3] Noci et al., *Signal propagation in transformers: Theoretical perspectives and the role of rank collapse.*
>
> [4] Liu et al., *Sophia: A Scalable Stochastic Second-order Optimizer for Language Model Pre-training.*
>
> [5] Fu et al., *Hungry Hungry Hippos: Towards Language Modeling with State Space Models.*
>
> [6] Dong et al, Attention is not all you need: Pure attention loses rank doubly exponentially with depth.

---

> ### Author Response · Authors · 2024-11-24
> **Rebuttal by Authors (2/2)**
>
> >Q1 Why are the target / smaller models initialized as k/2 ...
>
> Yes K is the number of total layers in the reference model (much larger vanilla models). We initialize target model with k/2 layers
> based on our empirical observation made in Fig.1 a, d and also newly added Fig. 10 that attention degeneration predominantly arises in later layers (beyond half way). Notably In a prior work [6] it is observed that in pure self attention networks without residual connections and Feed forward networks (FFNs) self-attention degenerates with depth (provably).
>
> Moreover in this paper we aim to match or outperform the reference model's val loss and hence starting from k/2 is reasonable to minimize number of rounds of training with Inheritune methodology. In Table 1 (original paper) row1 GPT-2 Medium (K=24 )we provide results with GPT2 variants initialized with 12 layers (K/2), 14 and 16 layers.
> Below we provide another ablation with GPT2 large here K=36.
>
> | Layer | Val Loss ↓ |
> |-------|------------|
> |   4   |    3.10    |
> |   8   |    2.89    |
> |  18   |    2.80    |
>
>
> New Table 1. Validation loss of GPT-2 Large variants as a function of initialized layers at the start of Inheritune (Algorithm 1).
>
> >Q2 Because you have the LLMs, some analysis into whether a layer is lazy or not ...
>
> In Fig. 1 we have plotted layerwise max ranks of vanilla GPT2 medium and GPT2 large models. The max rank indicates that which layers to retain during Inheritune training but these are always the initial layers. The max rank-1 means all the heads of a particular layer has been degenerated and this layer can be removed. Additionally we plotted rank analysis plot of GPT2-xLarge (https://postimg.cc/cKKJB1bB) also you may refer Fig. 10 in supplementary materials.
>
> >Q3 Is the step-count comparison fair?
>
> Yes it is fair. We have evaluated models trained with Inheritune in two settings: (1) larger models trained from scratch with same number of steps + same sized model trained from scratch for same and twice the number of training steps (Refer Table 1) . (2) Same sized models **not trained from scratch** with efficient training baselines etc (Refer Table 2).
>
> >Q4 Conversely, when making the claim that Inheritune ...
>
> No the larger models (reference models) in Table 1 are not further trained to match the number of steps. In Table 2 we have similar sizes models further trained with same number of steps following other baseline strategies of efficient model training.
>
> >Q5 In Figure 2, if the GPT2 models are decoder-only / causal, should the attention weights only be limited to lower-triangular?
>
> Yes, the attention maps of causal models should be lower triangular but as already mentioned in Sec. 2.1 for the sake of better visualization we have plotted the full attention matrix.

---

> > ### Comment · Reviewer_Bigz · 2024-11-25
> >
> > Thanks for your rebuttal. Appreciate the clarity on many of my initial questions and apologies for initial oversights.  I still have two remaining questions I think could help me better understand Inheritune's contribution.
> >
> > ### **Difference on lazy layers vs attention sinks**
> >
> > Thanks for clarifying this. So it seems like attention sinks are both a special case of lazy layers at the attention head level (when the rank-1 attention matrices end up displaying a dominant single-column structure (L080)), but also do not imply lazy layers (via your head-wise counterexample).
> >
> > * I am curious if you see any practical advantages to this distinction. i.e., do you encounter scenarios where layers are lazy but do not necessarily display attention sinks, motivating you to prune it where you wouldn't if you were just looking for attention sinks?
> >
> > ---
> >
> > Relatedly, it still seems a bit ad-hoc to me to initialize at $k/2$:
> >
> > >  We initialize target model with k/2 layers based on our empirical observation made in Fig.1 a, d and also newly added Fig. 10 that attention degeneration predominantly arises in later layers (beyond half way)
> >
> > * Given your observations + initial computation to identify lazy layers, why not base your method around a more granular set point that better connects to the lazy layer motivation?, e.g.,
> >
> > ---
> > Algorithm 1 suggested
> > 1. Given reference model $\mathcal{M}_\text{ref}$, identify latest contiguous non-lazy layer $n$  (or some other heuristic)
> > 2. Initialize target model $\mathcal{M}_\text{tgt}$ with first $n$ layers from M_ref
> > ...
> > ---
> > ---
> >
> >
> > ### **Limited model diversity (model family and scale)**
> >
> > Quoting author response:
> > > This is actually a great feedback. The attention maps are very sensitive to data distribution. Since we have trained the models with OpenWebText (OWT-9B) and then used the validation set to plot the attention maps we made sure that there is little to no distribution shift. This is very important for our setting because we are using rank-1 ness (attention degeneration) as a proxy for evaluating how well the layers are trained.
> > For the Gemma and Phi models, neither the training nor validation datasets are publicly available. Even if they were, we lack the computational resources required to train inheritune variants of these models, as they are trained on trillions of tokens.
> >
> > I guess what I also don't quite understand here (so please correct me if I'm wrong), is even without the pretrained data for Gemma or Phi, can you not just use some "probing data" (e.g., some tokens from OpenWebText validation set) to identify lazy layers in these models?
> > * Presumably, these closed-training-data models are trained with enough data s.t. OWT samples are within distribution? (at least enough that we should still be able to estimate the rank of the attention matrices). And if not, this sensitivity seems like a drawback to the method.
> >
> > Then, you could show that by using your approach, you can intelligently prune these models to num_layers // 2 (or using the suggested earlier approach of computing the latest non-lazy layer).
> > * where following the train + grow steps (where you only train over OWT or some token budget you do have access to), you can show you can recover similar performance to a drastically layer-reduced model, again via the specific pruning of lazy layers
> >
> > ---
> >
> > Again I like the lazy layer observation and think this is an interesting + well-motivated approach to model pruning. But with the current study I'm still left wondering about these questions and the method's impact (for wide-adoption with modern LLM research, can it support more recent models and be co-opted as a non-pretraining research method?)
> > * If the authors can address these questions (either with additional results, or responses on why they don't apply), I am happy to raise my score.
> >
> > ---
> >
> > ### **Additional nits**
> >
> > **Grammar** I ound a grammar error on L208,
> > > **How much transferable knowledge these lazy layers hold compared to their early counterparts?**
> >
> > should probably be
> > > **How much transferable knowledge *should* these lazy layers hold compared to their *earlier* counterparts?**
> >
> > **Citation style** For publication please use \citep{} for in-line parenthetical cites (e.g., L041-042):
> > >  However, as models grow in depth, they often encounter a phenomenon known as attention degeneration caused by collapse in the attention rank Noci et al. (2022); Dong et al. (2021); He et al. (2023).
> >
> > should be:
> > > However, as models grow in depth, they often encounter a phenomenon known as attention degeneration caused by collapse in the attention rank **(**Noci et al. (2022); Dong et al. (2021); He et al. (2023)**)**.
> >
> > **Missing reference** On L245,
> > > (also refer Figure ??)

---

> > > ### Author Response · Authors · 2024-11-28
> > > **Thanks for the discussion and liking our work. Response (1/2)**
> > >
> > > Thank you very much for all your thoughtful responses.
> > >
> > > > I am curious if you see any practical advantages to this distinction. i.e., do you encounter scenarios where layers are lazy but do not necessarily display attention sinks, motivating you to prune it where you wouldn't if you were just looking for attention sinks?
> > >
> > > Quoting myself from previous discussion.
> > >
> > > >The term "attention sink" [1] refers to the phenomenon where the first token in a sequence receives disproportionately high attention scores compared to other tokens in the attention maps.
> > >
> > > Attention sink is a general phenomenon and it occurs in attention matrices of lazy and non-lazy layers (refer current draft Figure 17) (also posted here: https://postimg.cc/FY7CPBF5). A lazy layer has all the attention matrices as attention sinks. However attention matrices of non-lazy layers may or may not have attention sinks. Pruning heads i.e. structural pruning is out of scope of this work and we defer this as a future work.
> > >
> > > > Relatedly, it still seems a bit ad-hoc to me to initialize at k/2 :
> > >
> > > In prior work [6], it was shown that self-attention in pure self-attention networks without residual connections or FFNs degenerates with depth. A key motivation for our work is to demonstrate that even with FFNs and residual connections, later layers can remain undertrained. Therefore, we opt to remove the later layers rather than the initial ones. Starting from layer \(k/2\) (halfway through the model) minimizes training rounds with the Inheritune methodology. We have already presented results for alternative starting points to \(k/2\) in previous responses.
> > >
> > > > Given your observations + initial computation to identify lazy layers, why not base your method around a more granular set point that better connects to the lazy layer motivation?, e.g.,
> > >
> > > > **Algorithm 1 suggested**
> > > > 1. Given reference model \( M_{\text{ref}} \), identify the latest contiguous non-lazy layer \( n \) (or some other heuristic).
> > > > 2. Initialize target model \( M_{\text{tgt}} \) with the first \( n \) layers from \( M_{\text{ref}} \).
> > >
> > > Thank you for the feedback. **We ran a small experiment with GPT2-small and it works**. We have added a discussion in supplementary Sec. D2 (also refer related Table 11 and Figure 15). Writing here for ease of reading. We trained a vanilla GPT2-small (125M) model with 12 layers for 100K steps using the OpenWebText (OWT-9B) dataset. First, we conducted a rank analysis of this model, as shown in Figure 15 (https://postimg.cc/NKcH5rfX).
> > >
> > > Next, we trained three GPT2-small variants for 100K steps, each with initialization from four contiguous layers of the vanilla GPT2-small model: (a) the first four layers [1, 2, 3, 4], (b) the middle four layers [5, 6, 7, 8], and (c) the last four layers [9, 10, 11, 12]. In addition, we trained another GPT2-small variant with random initialization. The key results are presented in Table 11 see below, and the complete training plots are shown in Figure 15 b) (https://postimg.cc/svSWjJyy). In summary, we observed that model initialized with layers from the vanilla GPT2 that exhibited higher max ranks (as indicated by Avg Max Ranks in table below) demonstrated the best performance.
> > >
> > >
> > > | Layers | Initialization                                   | Avg max ranks |  Val Loss ↓|
> > > |--------|----------------------------------------|--------------|----------|
> > > | 4      | rand                                   | n/a          | 3.25     |
> > > | 4      | 1-4 layers from vanilla GPT2          | 8.40         | 3.22     |
> > > | 4      | 5-8 layers from vanilla GPT2          | 9.48         | 3.19     |
> > > | 4      | 9-12 layers (lazy layers) from GPT2   | 1.22         | 3.23     |
> > >
> > > **New Table 1: 4 layer GPT2-small variant initialized with layers with higher max ranks achieves superior val loss (lower is better). Ablation of 4 layer GPT2-small model variants initialized with first four, middle four and last four layers (these are lazy layers) of vanilla GPT2-small model with 12 layers. All models are trained for 100K steps with OWT-9B data.
> > >
> > > In conclusion, we emphasize that when training a smaller Inheritune model to match or outperform the full-sized reference model, it is best to begin with the first half of the layers. However, if a specific target size is desired, selecting layers with the highest mean of maximum ranks can be more effective.

---

> ### Author Response · Authors · 2024-11-28
> **Response (2/2)**
>
> > Presumably, these closed-training-data models are trained with enough data s.t. OWT samples are within distribution? (at least enough that we should still be able to estimate the rank of the attention matrices). And if not, this sensitivity seems like a drawback to the method.
> > where following the train + grow steps (where you only train over OWT or some token budget you do have access to), you can show you can recover similar performance to a drastically layer-reduced model, again via the specific pruning of lazy layers
>
> ### Based on your feedback we conduced analysis with billion-sized LLMs.
>
> We conducted mass analysis (same way as discussed in Sec 2) for OpenLLaMA 3B (26 layers), 7B (32 layers) and 13B (40 layers) models. Refer Fig 16 in updated draft. Posting the Figure here as well (https://postimg.cc/0bbpy2Q6). We observe that 90% of the total mass of the attention matrices are concentrated in fewer columns i.e. in less than 5% of the total columns (i.e 100 for a 100x100 attention matrix) for 3B and 7B models. Next many of columns in later layers are single column for the 13B model.
>
> Next with Openllama 3B as reference model we developed a target model of 1.5B while only using 0.1% of the original pre-training tokens (only 1B tokens) whereas the reference model was trained using 1000B (1T) tokens. Refer supplementary Section B.
>
> ### We have addressed all the grammar and missing Figures reference issues as highlighted by you.

---

> > ### Comment · Reviewer_Bigz · 2024-11-28
> >
> > Thanks! I appreciate the extra experiments + follow-up. I've raised my score to 6.
> >
> > I still think the paper's presentation of the method could be improved, where I think you are proposing a layer-pruning method---and have a smart observation to motivate which layers to prune based on the lazy layers---but could better present the connections.
> > * Better to link the observation with the method directly; e.g., if GPT-2 with 12-layers has 8 lazy layers, start with the first 4 layers. If Llama 3 8B has 14 lazy layers, start with the first 18.
> > * This better isolates being able to exploit the lazy layer phenomenon, tightens the connection (whereas universally setting k // 2 feels like leaving some of the connection off the table), and opens up more interesting comparison points (if you prune based on lazy layers vs prune based on attention sinks or some other symptom of undertrained layers, is one more effective?)

---

> > > ### Author Response · Authors · 2024-12-01
> > > **Thank you for increasing the score and more discussion**
> > >
> > > Thanks for improving the score and sharing helpful feedback.
> > >
> > > We will try to improve our presentation and add more weight on the discussion about lazy layers. We already improved our analysis based our the previous discussion. Now we have mass analysis with large scale models trained on trillions of tokens.
> > >
> > > > Q1 Better to link the observation with the method directly; e.g., if GPT-2 with 12-layers has 8 lazy layers, start with the first 4 layers. If Llama 3 8B has 14 lazy layers, start with the first 18.
> > >
> > > This paper was not conceived as a pruning technique. Instead, it focuses on a structural issue in attention matrices. The key insight we aim to present is that the rank or mass of attention matrices can serve as a proxy for evaluating how effectively a layer has been trained. Next we show that bigger models with lazy layer (poorly trained later layers) can be replaced with smaller models trained with Inheritune without loosing any performance. Again this also a new insight.
> > >
> > > >Q2 This better isolates being able to exploit the lazy layer phenomenon, tightens the connection (whereas universally setting k // 2 feels like leaving some of the connection off the table), and opens up more interesting comparison points (if you prune based on lazy layers vs prune based on attention sinks or some other symptom of undertrained layers, is one more effective?)
> > >
> > > Throughout our discussions, we have conducted additional experiments using values other than \(k//2\). In the final draft, we will include a statement clarifying that \(k//2\) is a preferred hyperparameter but not a critical component of Inheritune. Additionally, in Figure 17, we have demonstrated that many non-lazy layers also exhibit the attention sink phenomenon. Whether these heads can be pruned requires further in-depth study.
> > >
> > > We sincerely appreciate your support for our work and would be truly grateful if you could share any suggestions for improvement that could lead to higher scores.

---

### Official Review · Reviewer_W6ma · 2024-11-04

**Soundness:** 3
**Presentation:** 3
**Contribution:** 2
**Rating:** 6
**Confidence:** 4

**Summary:**

The paper proposes a practical method "Inheritune" to create more efficient language models by reducing the layers of the original model. The authors analyze the phenomenon of "attention degeneration" in the deep layers of transformer language models and compare the proposed method against relevant baselines.

**Strengths:**

1. The analysis of attention degeneration phenomenon is well written and easy to understand. The motivation of this work is clear.
2. The paper proposes a practical and lightweight approach to perform model pruning that can be implemented with ease.

**Weaknesses:**

1. The scope of the degeneration analysis and experiments is limited. Only GPT-2 model variants have been discussed and evaluated. The generalization on other model architectures is unclear.
2. The choice of the initialization point n = k/2 assumes degeneration only takes place in the second half of the total number of layers, which is not supported by theoretical evidence.

**Questions:**

Please refer to Weaknesses section.

---

> ### Author Response · Authors · 2024-11-24
> **Author's Rebuttal**
>
> Thank you for the positive assessment of our work.
>
> >W1 The scope of the degeneration analysis and experiments is limited ...
>
> **Additional results : Mass Analysis of OpenLLaMA 3B, 7B and 13B trained with 1T tokens.**
>
> We conducted mass analysis (same way as discussed in Sec 2) for OpenLLaMA 3B (26 layers), 7B (32 layers) and 13B (40 layers) models. Refer Fig 16 in updated draft. Posting the Figure here as well (https://postimg.cc/0bbpy2Q6). We observe that 90% of the total mass of the attention matrices are concentrated in fewer columns i.e. in less than 5% of the total columns (i.e 100 for a 100x100 attention matrix) for 3B and 7B models. Next many of columns in later layers are single column for the 13B model.
>
> Next we also performed additional analysis with GPT2 model trained OpenWebText but the rank analysis os conducted with a new data Fineweb_Edu. See details below.
>
> In Fig 1 we have utilized vanilla GPT2 models trained with OpenwebText (OWT) and analysis was conducted using val set of OWT (refer Section 2). Next we took the same models trained with OWT but now we perform rank and mass analysis using new data Fineweb_edu( process remains the same). In New Figure (https://postimg.cc/4nPdR3yc) one can clearly see lazy layers in vanilla GPT2 medium , large and Xlarge models. Our analysis is quite robust. We will update the draft accordingly.
>
> >W2 The choice of the initialization point n = k/2 assumes degeneration only takes place
>
> The degeneration in attention matrices pre-dominantly occurs in the later layers [1]. This can also be thought as gradient vanishing in later layers [2]. Based on our rank analysis plots we have empirically observed that many of the later layers (beyond the half way mark) are lazy layers. This is an empirical observation and hence we don't have theoretical justification.
>
> Again thank you for reviewing our work.
>
> ---
> References
>
> [1] Dong et al, Attention is not all you need: Pure attention loses rank doubly exponentially with depth.
>
> [2] Barbero et al., Transformers need glasses! information over-squashing in language tasks.

---

> > ### Author Response · Authors · 2024-12-01
> > **Request to Respond**
> >
> > Dear Reviewer W6ma,
> >
> > Since we are very close to the end of the discussion period, please let us know if you have some additional comments or feedback about our work.

---

### Official Review · Reviewer_ZPxz · 2024-11-04

**Soundness:** 1
**Presentation:** 2
**Contribution:** 1
**Rating:** 3
**Confidence:** 5

**Summary:**

The paper introduces Inheritune, a training method designed to create smaller, high-performing language models. This method works by inheriting the initial layers from a larger pre-trained model and gradually expanding the smaller model until its performance matches or exceeds that of the larger model.

The authors investigate attention degeneration in standard large language models (LLMs) and find that rank-collapsed attention matrices often reduce to single-column structures, highlighting inefficiencies in the attention mechanism, especially in deeper layers. This insight into the limitations of deep attention inspired the development of Inheritune.

Experiments on different sizes of GPT-2 models, using the OpenWebText and FineWeb_Edu datasets, show that models trained with Inheritune, despite having significantly fewer layers, can match or even surpass the performance of larger models. They also outperform several baseline methods, such as stacking and knowledge distillation.

**Strengths:**

(+) The proposed training technique is simple and clear.
(+) The authors empirically investigate attention degeneration in standard LLM settings, focusing on rank collapse in attention matrices within the deeper layers of models, such as GPT-2.

**Weaknesses:**

(-) An analysis of whether attention degradation is resolved in the target model trained with Inheritune is missing and should be included.

(-) The adequacy of training for the reference model in Table 1 is uncertain. Given the hyperparameter setting, the GPT2-large (770M) model was trained with only 1.6 billion tokens, using a batch size of 16K tokens and a total of 100K steps. This training setup appears insufficient for comprehensive model training. Thus, it is crucial to compare the performance of the target model against a reference model that has undergone adequate training.

(-) Since the Inheritune target model begins with the weights of the pretrained model, it has a much lower initial loss at step 0, giving it a head start compared to models trained from scratch. As a result, it’s difficult to determine whether models using Inheritune truly lead to better generalization and convergence.

(-) The baselines used for comparison need to be updated to reflect advancements in training techniques. Some of the latest research on half-width models, stacking, and distillation that would serve as recommended baselines are as follows:
•	Xiaoqi Jiao et el., TinyBERT: Distilling BERT for Natural Language Understanding, 2020
•	Sheng Shen et al., Staged Training for Transformer Language Models, 2022
•	Peihao Wang et el., LEARNING TO GROW PRETRAINED MODELS FOR EFFICIENT TRANSFORMER TRAINING, 2023

(-) There seems to be a typo: line 323, "layers 0-17" should be "layers 9-17."

**Questions:**

1. Why are the blanks for downstream task performance in Table 1 not filled in?
2. According to the training details of the GPT-2 models (Supplement C.1), it appears that GPT-2 medium was trained on more data than GPT-2 xLarge. Specifically, GPT-2 medium was trained with a batch size of 50K tokens for 100K steps (totaling 5 billion tokens), while GPT-2 xLarge used a batch size of only 16K tokens for the same number of steps (totaling 1.6 billion tokens). Is this a typo?
3. Have you considered initializing the layers added during the growth phase using methods such as random initialization or copying previous layers of the target model? This is because the layers added during the growth phase closely resemble the "lazy layers" described in the paper.
4. It would be helpful to provide a clear criterion for identifying a specific layer as a lazy layer (e.g., based on the rank, mass, or other properties of the attention matrix).

---

> ### Author Response · Authors · 2024-11-28
> **Rebuttal by Authors**
>
> > W1 An analysis of whether attention degradation is resolved ...
>
> The degradation is definitely mitigated compared to the full sized reference model. Please refer Figure 9 and Fig 10 of the updated draft. Also posting the figures here (https://postimg.cc/bGKwCcXc) and (https://postimg.cc/cKKJB1bB).
>
> > W2 The adequacy of training for the reference model ...
>
> We have inherited the pre-training set up from [2]v1. For pre-training experiments in many prior works such as [2],[3],[4], and [5] GPT2 models are trained for 100K steps using OpenWebText (OWT-9B). Moreover in one of our baseline in Table 1 we have trained our baseline GPT2 variants from scratch for 200K steps. We emphasis that our recipe works well on GPT2-medium, GPT2-large and GPT2-xLarge variants and this provides a stronger signal of the efficacy of our method.
>
> **Additional analysis with Billion sized LLMs**
>
> We conducted mass analysis (same way as discussed in Sec 2) for OpenLLaMA 3B (26 layers), 7B (32 layers) and 13B (40 layers) models. Refer Fig 16 in updated draft. Posting the Figure here as well (https://postimg.cc/0bbpy2Q6). We observe that 90% of the total mass of the attention matrices are concentrated in fewer columns i.e. in less than 5% of the total columns (i.e 100 for a 100x100 attention matrix) for 3B and 7B models. Next many of columns in later layers are single column for the 13B model.
>
> With Openllama 3B as reference model we developed a target model of 1.5B while only using 0.1% of the original pre-training tokens (only 1B tokens) whereas the reference model was trained using 1000B (1T) tokens. Refer supplementary Section B.
>
> > W3 Since the Inheritune target model begins with the weights of the pretrained model, it has a much lower initial loss at step 0 ...
>
> **The main claim of this paper is not that models trained with Inheritune beats models trained from scratch.**  We have evaluated models trained with Inheritune in two settings: (1) larger models trained from scratch with same number of steps + same sized model trained from scratch for same and twice the number of training steps (Refer Table 1) . (2) Same sized models **not trained from scratch** with efficient training baselines etc (Refer Table 2). Therefore we believe that the comparison is fair.
>
> Additionally in any prior language modelling literature (LLM pre-training) is was not known that a much smaller model can match or outperform larger model in pre-train val loss (lower is better). We believe this is happening cause larger LLMs suffers from lazy layers. This insight in novel and can be helpful to practitioners. Due to academic compute budget we conducted our experiments with limited number of training steps.
>
> > W4 The baselines used for comparison need to be updated to reflect advancements in training techniques ...
>
> In Table 2, we compared against variants of stacking [6] [7] these works are published in 2023 and 2019 and are known efficient training recipes. Next We have we have also compared again half-width and hybrid stacking that we developed as a baseline to be fair as stacking uses it's own weights and hybrid stacking used weights of a larger reference model. Finally we also compared against well know Distill BERT style distillation and initialization with teacher layer [8] (refer Figure 3 and 8).
>
> In a prior work [1] it is observed that in pure self attention networks without residual connections and Feed forward networks (FFNs) attention degenerates with depth (provably). An important contribution of our paper is to show that even with residual connections and FFNs attention may loose rank as shown empirically in Figure 1. Using Inheritune, we showed that it is possible to remove lazy layers, re-train the model, and selectively include a few lazy layers during the growth stage to match or even outperform the performance of the full-sized model. We are curious, if you may please clarify why do you feel that the current baselines are in-sufficient and how papers suggested by you such as TinyBERT and growth operator based training are relevant to this work? As we believe non of these works are solving the problem discussed above. Additionally we have already compared against Distill BERT (refer Figure 3 and 8).
>
> > W5 There seems to be a typo ...
>
> It's not a typo. In hybrid stacking we copy 0-8 layers from reference model's layers to 0-8 and 9-17 layers of the target model.

---

> > ### Author Response · Authors · 2024-11-28
> > **Rebuttal by Authors (2/2)**
> >
> > >Q1 Why are the blanks for downstream task ...
> >
> > We only evaluate downstream performance final models and model trained for 100K steps (i.e. same # of steps w.r.t ours).
> >
> > > Q2 According to the training details of the GPT-2 models (Supplement C.1), it appears that GPT-2 medium was trained on more data than GPT-2 xLarge ...
> >
> > GPT2 medium models trained with OWT are indeed trained with more data. This is because we 16K is the largest batch size fits our memory budget. This is due the usage of limited academic resources.
> >
> > > Q3 Have you considered initializing the layers added during the growth phase using methods such as random initialization or copying previous layers of the target model? This is because the layers added during the growth phase closely resemble the "lazy layers" described in the paper.
> >
> > You mentioned "copying previous layers of the target model" -- Yes the methodology you described is stacking.
> >
> > You mentioned "as random initialization" -- Not exactly random initialization but in Inheritune we intentionally use lazy layers during growth phase and this resembles close to random init as shown in Figure 1. Again yes we use lazy layers in growth phase but now the important distinction is we are training an overall smaller model so, we hope to train every layer better.
> >
> > > Q4 It would be helpful to provide a clear criterion for identifying a specific layer as a lazy layer ...
> >
> > The rank analysis, mass analysis, and training curve resembling random initialization in Figure 1 all point to the conclusion that the later layers are poorly trained. A significant prior work [1] demonstrated that in pure self-attention networks without residual connections or feedforward networks (FFNs), attention provably degenerates with depth. Building on this, our paper makes the important contribution of showing that even with residual connections and FFNs, attention still degenerates. While we measure this phenomenon using rank and mass analysis, the underlying issue is poor training of later layers. Empirically, we observed that attention degradation becomes more prominent beyond the halfway point of the network, which is why we prioritize the use of early layers.
> >
> >
> > **References**
> >
> > [1] Dong et al, Attention is not all you need: Pure attention loses rank doubly exponentially with depth.
> >
> > [2] Liu et al., Sophia: A Scalable Stochastic Second-order Optimizer for Language Model Pre-training.
> >
> > [3] Sanyal et al., Early Weight Averaging meets High Learning Rates for LLM Pre-training.
> >
> > [4] Taniguchi et al., ADOPT: Modified AdamCanConverge with Any β2 with the Optimal Rate.
> >
> > [5] Fu et al., Hungry Hungry Hippos: Towards Language Modeling with State Space Models.
> >
> > [6] Reddi et al., Efficient training of language models using few-shot learning.
> >
> > [7] Gong et al., Efficient training of BERT by progressively stacking.
> >
> > [8] Sanh et al., Distilbert,a distilled version of bert: smaller, faster, cheaper and lighter.

---

> ### Author Response · Authors · 2024-12-01
> **Request to Respond by Authors: Provided clarifications with additional experiments**
>
> Dear Reviewer ZPxz,
>
> Thank you for your thoughtful feedback. We have provided additional clarifications and results to demonstrate how Inheritune mitigates attention degeneration. Furthermore, we have included a mass analysis using OpenLLaMA models (3B, 7B, and 13B) trained on 1T tokens. If you feel that we have sufficiently addressed your concerns, we kindly request you to consider revisiting and potentially improving your scores.

---

> > ### Author Response · Authors · 2024-12-02
> > **Request to respond by Authors**
> >
> > Dear Reviewer ZPxz,
> >
> > Thanks for taking some time to review our work. We feel your thoughtful remarks have improved the quality of our work. Since we are close to end of discussion period we would love to get some additional feedback on the comments we made during the rebuttal. We also invite you to update your scores only if you believe we have sufficiently addressed your concerns.

---

### Author Response · Authors · 2024-11-28
**General Response**

We sincerely thank all the reviewers for taking the time to review our work. Below, we summarize the major issues raised by the reviewers and provide our general responses to address them.

We have added a new supplementary section  Section D where we have registered new results and discussions done during the rebuttal phase. Additionally Figure 10 which is not a part of Section D is also added during rebuttal.

> An analysis of whether attention degradation is resolved in the target model trained with Inheritune is missing and should be included.

The degradation is definitely mitigated compared to the full sized reference model. Please refer Figure 9 and Fig 10 of the updated draft. Also posting the figures here (https://postimg.cc/bGKwCcXc) and (https://postimg.cc/cKKJB1bB).

> The attention degradation is observed in GPT2 models which may be undertrained?

We conducted mass analysis (same way as discussed in Sec 2) for OpenLLaMA 3B (26 layers), 7B (32 layers) and 13B (40 layers) models. Refer Fig 16 in updated draft. Posting the Figure here as well (https://postimg.cc/0bbpy2Q6). We observe that 90% of the total mass of the attention matrices are concentrated in fewer columns i.e. in less than 5% of the total columns (i.e 100 for a 100x100 attention matrix) for 3B and 7B models. Next many of attention matrices in later layers are single column for the 13B model.

> Lazy layers are similar to Attention Sinks.

The term "attention sink" [1] refers to the phenomenon where the first token in a sequence receives disproportionately high attention scores compared to other tokens in the attention maps. While there is some connection, as we have also observed that many attention matrices are not only rank-1 but also single-column (with all attention scores concentrated on the first token), this connection has not been explicitly established in [1] with respect to rank-1 behavior or poor training of later layers.

In contrast, as illustrated in Figure 1 of our paper, we compute the maximum rank (plotted on the y-axis) of all attention matrices within a layer. We define a layer as "lazy" only if the maximum rank of its attention matrices is 1, as discussed in Section 2. For instance, consider a layer where only 2 out of 5 attention heads exhibit attention sink behavior. This does not make the layer lazy, as attention is computed as a concatenation of activations across all heads. A truly lazy layer, however, has all 5 out of 5 attention heads fully degenerated, with their attention matrices being rank-1. We provide evidence that such lazy layers are indicative of poorly trained layers.

Attention sink is a general phenomenon and it occurs in attention matrices of lazy and non-lazy layers (refer current draft Figure 17) (also posted here: https://postimg.cc/FY7CPBF5). A lazy layer has all the attention matrices as attention sinks. However attention matrices of non-lazy layers may or may not have attention sinks. Pruning heads i.e. structural pruning is out of scope of this work and we defer this as a future work.


> Limited Model diversity in experiments.

We have demonstrated that Inheritune is effective across various scales of GPT-2 models, including GPT-2 Medium (355M), GPT-2 Large (770M), and GPT-2 XL (1.5B), all trained for 100K steps on the OpenWebText dataset. Additionally, we trained GPT-2 Medium* and GPT-2 Large models on the fineweb_edu dataset to further validate our approach. Moreover, we showed the efficacy of Inheritune on the OpenLLaMA-1.5B model variant, which was trained on 1B tokens. These results establish the broad applicability of Inheritune across diverse model architectures and datasets. Unfortunately, due to constraints in our academic compute budget, we were unable to train the models for longer durations.

> Does the lazy layer hypothesis holds when the analysis is performed using a different data?

We performed additional experiments. In Fig 1 we have utilized vanilla GPT2 models trained with OpenwebText (OWT) and analysis was conducted using val set of OWT (refer Section 2). Next we took the same models trained with OWT but now we perform rank and mass analysis using **new data Fineweb_edu**( process remains the same). In New Figure (https://postimg.cc/4nPdR3yc) one can clearly see lazy layers in vanilla GPT2 medium , large and Xlarge models. Our analysis is quite robust. We will update the draft accordingly.

---

### Author Response · Authors · 2024-12-02
**Summary of Discussions during Rebuttal**

We thank the reviewers for their feedback.

- **Reviewers Bigz and FevH** (2/4) actively participated in the rebuttal discussions and subsequently increased their evaluation scores. Both reviewers find this work promising and impactful.

- **Reviewer W6ma** acknowledged the strong motivation behind this work. Concerns regarding the lack of model diversity (W1) were addressed during the rebuttal process.

### Key Updates

1. **Extended Analysis to Larger Models**
   We expanded the attention degeneration analysis to larger models, including OpenLLaMA 3B, 7B, and 13B, as shown in Figure 16. (Discussion with Reviewer Bigz)

2. **Robustness to Data Proven**
   Through additional analysis, we demonstrated that our attention degeneration analysis is robust across datasets. Refer to Figure 18 ([link](https://postimg.cc/4nPdR3yc)), which will be updated in the final draft. (Discussion with Reviewer FevH)

3. **Mitigation of Attention Degeneration via Inheritune**
   We provided clear evidence that Inheritune effectively mitigates attention degeneration. See Figures 9 and 10. (Discussion with Reviewer Zpxz)

4. **Correlation with Generalization**
   We established that layers with higher average maximum ranks of attention matrices correlate with better generalization of target model. Refer to Figure 15.  (Discussion with Reviewer FevH)

Overall, none of the reviewers have concerns about the novelty of this method, and the majority of reviewers (3/4) had their concerns about the generality of the attention degeneration analysis and its mitigation through Inheritune addressed during the rebuttal.

---

### Meta-Review · Area_Chair_tnMe · 2024-12-20

**Metareview:**

The paper proposes a novel training paradigm aimed at improving the efficiency of language models by reducing their size while maintaining or enhancing their performance. The key idea is to use a technique termed "Inheritune," which involves selectively fine-tuning smaller models to inherit attention patterns from larger pre-trained models. This approach leads to smaller models that are more computationally efficient and attentive, achieving competitive results on multiple benchmarks. Experimental results are provided to demonstrate the effectiveness of the method, with comparisons to baseline techniques such as knowledge distillation and other model compression strategies.

The paper introduces "Inheritune," which is an innovative approach to training smaller language models by transferring attention patterns from larger models. This demonstrates creativity and relevance to the field. It tries to address a critical problem in NLP, reducing the computational cost of large-scale language models. However, the paper lacks a strong theoretical foundation to explain why "Inheritune" works better than existing methods like knowledge distillation. This limits confidence in its broader applicability. The experimental evaluation is not comprehensive enough, as it does not include comparisons with all relevant state-of-the-art baselines or tests on sufficiently diverse datasets. With these concerns, I recommend rejection for this submission. I hope the authors can improve the paper and submit it to a future venue.

**Additional Comments On Reviewer Discussion:**

## Points Raised by Reviewers:
1. Theoretical Justification: Reviewers questioned why "Inheritune" should work better than other techniques like knowledge distillation and requested more theoretical insights.
2. Experimental Baselines: Concerns were raised about missing comparisons with stronger baselines and state-of-the-art methods.
Reproducibility: Several reviewers flagged incomplete implementation details as a barrier to reproducibility.
3. Significance of Results: Some reviewers felt that the performance gains were incremental and did not justify the novelty claims.
## Authors’ Responses
1. Theoretical Justification: The authors acknowledged this limitation but argued that their empirical results provided sufficient evidence of effectiveness.
2. Experimental Baselines: They added additional comparisons during the rebuttal period but could not include all requested baselines due to time constraints.
3. Significance of Results: The authors emphasized that their method targets efficiency improvements rather than absolute performance gains.

## Final Decision Considerations
Each point was carefully weighed in arriving at the final decision: The lack of theoretical justification remained a significant weakness since it undermines confidence in generalizability beyond specific benchmarks. While additional baselines were included during rebuttal, they were insufficient to address all reviewer concerns about experimental rigor. Incremental gains in performance were not deemed sufficient to justify acceptance given the high standards for novelty and impact at this conference.

---

### Decision · Program_Chairs · 2025-01-22

Reject